# Accelerate Multi-Agent Reinforcement Learning in Zero-Sum Games with Subgame Curriculum Learning

## Abstract

Learning Nash equilibrium (NE) in complex zero-sum games with multi-agent reinforcement learning (MARL) can be extremely computationally expensive. Curriculum learning is an effective way to accelerate learning, but an under-explored dimension for generating a curriculum is the difficulty-to-learn of the *subgames* – games induced by starting from a specific state. In this work, we present a novel subgame curriculum learning framework for zero-sum games. It adopts an adaptive initial state distribution by resetting agents to some previously visited states where they can quickly learn to improve performance. Building upon this framework, we derive a subgame selection metric that approximates the squared distance to NE values and further adopt a particle-based state sampler for subgame generation. Integrating these techniques leads to our new algorithm, *Subgame Automatic Curriculum Learning* (SACL), which is a realization of the subgame curriculum learning framework. SACL can be combined with any MARL algorithm such as MAPPO. Experiments in the particle-world environment and Google Research Football environment show SACL produces much stronger policies than baselines. In the challenging hide-and-seek quadrant environment, SACL produces all four emergent stages and uses only half the samples of MAPPO with self-play. The project website is at https://sites.google.com/view/sacl-neurips.

## 1 Introduction

Applying reinforcement learning (RL) to zero-sum games has led to enormous success, with trained agents defeating professional humans in Go [41], StarCraft II [46], and Dota 2 [5]. To find an approximate Nash equilibrium (NE) in complex games, these works often require a tremendous amount of training resources including hundreds of GPUs and weeks or even months of time. The unaffordable cost prevents RL from more real-world applications beyond these flagship projects supported by big companies and makes it important to develop algorithms that can learn close-to-equilibrium strategies in a substantially more efficient manner.

One way to accelerate training is curriculum learning – training agents in tasks from easy to hard. Many existing works in solving zero-sum games with MARL generate a curriculum by choosing whom to play with. They often use self-play to provide a natural policy curriculum as the agents are trained against increasingly stronger opponents [4, 2]. The self-play framework can be further extended to population-based training (PBT) by maintaining a policy pool and iteratively training new best responses to mixtures of previous policies [31, 23]. Such a policy-level curriculum generation paradigm is very different from the paradigm commonly used in goal-conditioned RL [29, 35]. Most curriculum learning methods for goal-conditioned problems directly reset the goal or initial states for each training episode to ensure the current task is of suitable difficulty for the learning agent. In contrast, the policy-level curriculum in zero-sum games only provides increasingly stronger

opponents, and the agents are still trained by playing the full game starting from a fixed initial state distribution, which is often very challenging.

In this paper, we propose a general subgame curriculum learning framework to further accelerate MARL training for zero-sum games. It leverages ideas from goal-conditioned RL. Complementary to policy-level curriculum methods like self-play and PBT, our framework generates subgames (i.e., games induced by starting from a specific state) with growing difficulty for agents to learn and eventually solve the full game. We provide justifications for our proposal by analyzing a simple iterated Rock-Paper-Scissors game. We show that in this game, vanilla MARL requires exponentially many samples to learn the NE. However, by using a buffer to store the visited states and choosing an adaptive order of state-induced subgames to learn, the NE can be learned with linear samples.

A key challenge in our framework is to choose which subgame to train on. This is non-trivial in zero-sum games since there does not exist a clear progression metric like the success rate in goal-conditioned problems. While the squared difference between the current state value and the NE value can measure the progress of learning, it is impossible to calculate this value during training as the NE is generally unknown. We derive an alternative metric that approximates the squared difference with a bias term and a variance term. The bias term measures how fast the state value changes and the variance term measures how uncertain the current value is. We use the combination of the two terms as the sampling weights for states and prioritize subgames with fast change and high uncertainty.

Instantiating our framework with the state selection metric and a non-parametric subgame sampler, we develop an automatic curriculum learning algorithm for zero-sum games, i.e., *Subgame Automatic Curriculum Learning* (SACL). SACL can adopt any MARL algorithm as its backbone and preserve the overall convergence property. In our implementation, we choose the MAPPO algorithm [52] for the best empirical performances.

We first evaluate SACL in the Multi-Agent Particle Environment and Google Research Football, where SACL learns stronger policies with lower exploitability than existing MARL algorithms for zero-sum games given the same amount of environment interactions. We then stress-test the efficiency of SACL in the challenging hide-and-seek environment. SACL leads to the emergence of all four phases of different strategies and uses 50% fewer samples than MAPPO with self-play.

## 2   Preliminary

### 2.1   Markov game

A Markov game [25] is defined by a tuple $\mathcal{MG} = (\mathcal{N}, \mathcal{S}, \boldsymbol{\mathcal{A}}, P, \boldsymbol{R}, \gamma, \rho)$, where $\mathcal{N} = \{1, 2, \cdots, N\}$ is the set of agents, $\mathcal{S}$ is the state space, $\boldsymbol{\mathcal{A}} = \Pi_{i=1}^{N} \mathcal{A}_i$ is the joint action space with $\mathcal{A}_i$ being the action space of agent $i$, $P : \mathcal{S} \times \boldsymbol{\mathcal{A}} \to \Delta(\mathcal{S})$ is the transition probability function, $\boldsymbol{R} = (R_1, R_2, \cdots, R_N) :$ $\mathcal{S} \times \boldsymbol{\mathcal{A}} \to \mathbb{R}^n$ is the joint reward function with $R_i$ being the reward function for agent $i$, $\gamma$ is the discount factor, and $\rho$ is the distribution of initial states. Given the current state $s$ and the joint action $\boldsymbol{a} = (a_1, a_2, \cdots, a_N)$ of all agents, the game moves to the next state $s'$ with probability $P(s'|s, \boldsymbol{a})$ and agent $i$ receives a reward $R_i(s, \boldsymbol{a})$.

For infinite-horizon Markov games, a subgame $\mathcal{MG}(s)$ is defined as the Markov game induced by starting from state $s$, i.e., $\rho(s) = 1$. Selecting subgames is therefore equivalent to setting the Markov game's initial states. The subgames of finite-horizon Markov games are defined similarly and have an additional variable to denote the current step $t$.

We focus on two-player zero-sum Markov games, i.e., $N = 2$ and $R_1(s, \boldsymbol{a}) + R_2(s, \boldsymbol{a}) = 0$ for all state-action pairs $(s, \boldsymbol{a}) \in \mathcal{S} \times \boldsymbol{\mathcal{A}}$. We use the subscript $i$ to denote variables of player $i$ and the subscript $-i$ to denote variables of the player other than $i$. Each player uses a policy $\pi_i : \mathcal{S} \to \mathcal{A}_i$ to produce actions and maximize its own accumulated reward. Given the joint policy $\boldsymbol{\pi} = (\pi_1, \pi_2)$, each player's value function of state $s$ and Q-function of state-action pair $(s, \boldsymbol{a})$ are defined as

$$V_i^{\boldsymbol{\pi}}(s) = \mathbb{E}_{\boldsymbol{a}^t \sim \boldsymbol{\pi}(\cdot|s^t), s^{t+1} \sim P(\cdot|s^t, \boldsymbol{a}^t)} \Big[ \sum_t \gamma^t R_i(s^t, \boldsymbol{a}^t) \Big| s^0 = s \Big], \tag{1}$$

$$Q_i^{\boldsymbol{\pi}}(s, \boldsymbol{a}) = \mathbb{E}_{\boldsymbol{a}^t \sim \boldsymbol{\pi}(\cdot|s^t), s^{t+1} \sim P(\cdot|s^t, \boldsymbol{a}^t)} \Big[ \sum_t \gamma^t R_i(s^t, \boldsymbol{a}^t) \Big| s^0 = s, \boldsymbol{a}^0 = \boldsymbol{a} \Big]. \tag{2}$$

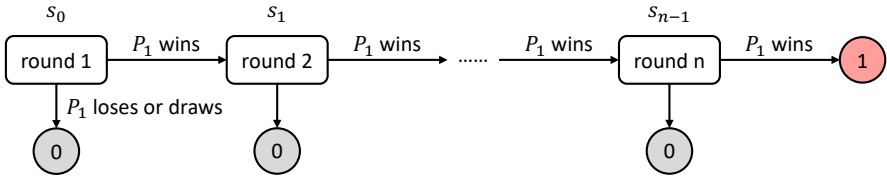

Figure 1: Illustration of the iterated Rock-Paper-Scissors game.

The solution concept of two-player zero-sum Markov games is Nash equilibrium (NE), which is a joint policy where no player can get a higher value by changing its policy alone.

**Definition 1** (NE). *A joint policy $\boldsymbol{\pi}^* = (\pi_1^*, \pi_2^*)$ is a Nash equilibrium of a Markov game if for all initial states $s^0$ with $\rho(s^0) > 0$, the following condition holds*

$$\pi_i^* = \arg\max_{\pi_i} V_i^{(\pi_i, \pi_{-i}^*)}(s^0), \ \forall i \in \{1, 2\}. \tag{3}$$

We use $V_i^*(\cdot)$ to denote the NE value function of player $i$ and $Q_i^*(\cdot, \cdot)$ to denote the NE Q-function of player $i$, and the following equations hold by definition and the minimax nature of zero-sum games.

$$V_i^*(s) = \max_{\pi_i} \min_{\pi_{-i}} \mathbb{E}_{\boldsymbol{a} \sim \boldsymbol{\pi}(\cdot|s)} \left[ Q_i^*(s, \boldsymbol{a}) \right], \tag{4}$$

$$Q_i^*(s, \boldsymbol{a}) = R_i(s, \boldsymbol{a}) + \gamma \cdot \mathbb{E}_{s' \sim P(\cdot|s, \boldsymbol{a})} \left[ V_i^*(s') \right]. \tag{5}$$

## 2.2 MARL algorithms in zero-sum games

MARL methods have been applied to zero-sum games tracing back to the TD-Gammon project [45]. A large body of work [54, 6, 42, 16] is based on regret minimization, and a well-known result is that the average of policies produced by self-play of regret-minimizing algorithms converges to the NE policy of zero-sum games [15]. Another notable line of work [25, 17, 23, 34] combines RL algorithms with game-theoretic approaches. These works typically use self-play or population-based training to collect samples and then apply RL methods like Q-learning [51] and PPO [39] to learn the NE value functions and policies, and have recently achieved great success [41, 20, 46, 5].

For the analysis in the next section, we introduce a classic MARL algorithm named minimax-Q learning [25] that extends Q-learning to zero-sum games. Initializing functions $Q_i(\cdot, \cdot)$ with zero values, minimax-Q uses an exploration policy induced by the current Q-functions to collect a batch of samples $\{(s^t, \boldsymbol{a}^t, r_i^t, s^{t+1})\}_{t=0}^T$ and uses these samples to update the Q-functions by

$$Q_i(s^t, \boldsymbol{a}^t) \leftarrow (1 - \alpha) \cdot Q_i(s^t, \boldsymbol{a}^t) + \alpha \cdot \left( r_i^t + \gamma \cdot \max_{\pi_i} \min_{\pi_{-i}} \mathbb{E}_{\boldsymbol{a} \sim \boldsymbol{\pi}(\cdot|s)} \left[ Q_i(s^{t+1}, \boldsymbol{a}) \right] \right), \tag{6}$$

where $\alpha$ is the learning rate. This sample-and-update process continues until the Q-functions converge. Under the assumptions that the state-action sets are discrete and finite and are visited an infinite number of times, it is proved that the stochastic updates by Eq. (6) leads to the NE Q-functions [43].

# 3 A motivating example

In this section, we show by a simple illustrative example that vanilla MARL methods like minimax-Q require exponentially many samples to derive the NE. However, if we can dynamically set the initial state distribution and induce an appropriate order of subgames to learn, the sample complexity can be substantially reduced from exponential to linear. Such an observation motivates our proposed algorithm described in later sections.

## 3.1 Iterated Rock-Paper-Scissors game

We introduce an iterated variant of the Rock-Paper-Scissor (RPS) game, denoted as $RPS(n)$. As shown in Fig. 1, $P_1$ and $P_2$ play the RPS game for up to $n$ rounds. If $P_1$ wins all rounds, it gets a reward of 1 and $P_2$ gets a reward of $-1$. If $P_1$ loses or draws in any round, the game ends immediately without playing the remaining rounds and both players get zero rewards. Note that the $RPS(n)$ game

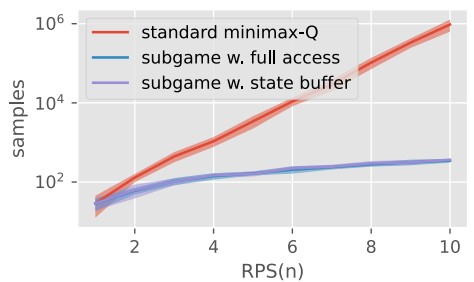

Figure 2: Number of samples used to learn the NE Q-values of $RPS(n)$ games.

**Algorithm 1:** Subgame curriculum learning

**Input:** state sampler $\mathrm{oracle}(\cdot)$.
Initialize policy $\boldsymbol{\pi}$;
**repeat**
 Sample $s^0 \sim \mathrm{oracle}(\mathcal{S})$;
 Rollout $\boldsymbol{\pi}$ in $\mathcal{MG}(s^0)$;
 Train $\boldsymbol{\pi}$ via MARL;
**until** $\boldsymbol{\pi}$ *converges*;
**Output:** final policy $\boldsymbol{\pi}$.

is different from playing the RPS game repeatedly for $n$ times because players can play less than $n$ rounds and they only receive a non-zero reward if $P_1$ wins in all rounds. We use $s_k$ to denote the state where players have already played $k$ RPS games and are at the $k+1$ round. It is easy to verify that the NE policy for both players is to play Rock, Paper, or Scissors with equal probability at each state. Under this joint NE policy, $P_1$ can win one RPS game with $1/3$ probability, and the probability for $P_1$ to win all $n$ rounds and get a non-zero reward is $1/3^n$.

Consider using standard minimax-Q learning to solve the $RPS(n)$ game. With Q-functions initialized to zero, we execute the exploration policy to collect samples and perform the update in Eq. (6). Note all state-actions pairs are required to be visited to guarantee convergence to the NE. Therefore, in this sparse-reward game, random exploration will clearly take $\mathcal{O}(3^n)$ steps to get a non-zero reward. Moreover, even if the exploration policy is perfectly set to the NE policy of $RPS(n)$, the probability for $P_1$ to get the non-zero reward by winning all RPS games is still $\mathcal{O}(1/3^n)$, requiring at least $\mathcal{O}(3^n)$ samples to learn the NE Q-values of the $RPS(n)$ game.

### 3.2 From exponential to linear complexity

An important observation is that the states in later rounds become exponentially rare in the samples generated by starting from the fixed initial state. If we can directly reset the game to these states and design a smart order of minimax-Q updates on the subgames induced by these states, the NE learning can be accelerated significantly. Note that $RPS(n)$ can be regarded as the composition of $n$ individual $RPS(1)$ games, a suitable order of learning would be from the easiest subgame $RPS(1)$ starting from state $s_{n-1}$ to the full game $RPS(n)$ starting from state $s_0$. Assuming we have full access to the state space, we first reset the game to $s_{n-1}$ and use minimax-Q to solve subgame $RPS(1)$ with $\mathcal{O}(1)$ samples. Given that the NE Q-values of $RPS(k)$ are learned, the next subgame $RPS(k+1)$ is equivalent to an $RPS(1)$ game where the winning reward is the value of state $s_{n-k}$. By sequentially applying minimax-Q to solve all $n$ subgames from $RPS(1)$ to $RPS(n)$, the number of samples required to learn the NE Q-values is reduced substantially from $\mathcal{O}(3^n)$ to $\mathcal{O}(n)$.

In practice, we usually do not have access to the entire state space and cannot directly start from the last subgame $RPS(1)$. Instead, we can use a buffer to store all visited states and gradually span the state space. By resetting games to the newly visited states, the number of samples required to cover the full state space is still $\mathcal{O}(n)$, and we can then apply minimax-Q from $RPS(1)$ to $RPS(n)$. Therefore, the total number of samples is still $\mathcal{O}(n)$. The detailed analysis can be found in Appendix A.1. We validate our analysis by running experiments on $RPS(n)$ games for $n = 1, \cdots, 10$ and the results averaged over ten seeds are shown in Fig. 2. It can be seen that the sample complexity reduces from exponential to linear by running minimax-Q over a smart order of subgames, and the result of using a state buffer in practice is comparable to the result with full access.

## 4 Method

The motivating example suggests that NE learning can be largely accelerated by running MARL algorithms in a smart order over states. Inspired by this insight, we present a general framework to accelerate NE learning in zero-sum games by training over a curriculum of subgames. We further propose two practical techniques to instantiate the framework and present the overall algorithm.

## 4.1 Subgame curriculum learning

The key issue of the standard sample-and-update framework is that the rollout trajectories always start from the fixed initial state distribution $\rho$, so visiting states that are most critical for efficient learning can consume a large number of samples. To accelerate training, we can directly reset the environment to those critical states. Suppose we have an oracle state sampler $\text{oracle}(\cdot)$ that can initiate suitable states for the current policy to learn, i.e., generate appropriate induced subgames, we can derive a general-purpose framework in Alg. 1, which we call subgame curriculum learning. Note that this framework is compatible with any MARL algorithm for zero-sum Markov games.

A desirable feature of subgame curriculum learning is that it does not change the convergence property of the backbone MARL algorithm, as discussed below.

**Proposition 1.** *If all initial states $s^0$ with $\rho(s^0) > 0$ are sampled infinitely often, and the backbone MARL algorithm is guaranteed to converge to an NE in zero-sum Markov games, then subgame curriculum learning also produces an NE of the original Markov game.*

The proof can be found in Appendix A.2. Note that such a requirement is easy to satisfy. For example, given any state sampler $\text{oracle}(\cdot)$, we can construct a valid mixed sampler by sampling from $\text{oracle}(\cdot)$ for probability $0 < p < 1$ and sampling from $\rho$ for probability $1 - p$.

**Remark.** With a given state sampler, the only requirement of our subgame curriculum learning framework is that the environment can be reset to a desired state to generate the induced game. This is a standard assumption in the curriculum learning literature [13, 29, 35] and is feasible in many RL environments. For environments that do not support this feature, we can simply reimplement the reset function to make them compatible with our framework.

## 4.2 Subgame sampling metric

A key question is how to instantiate the oracle sampler, i.e., *which subgame should we train on for faster convergence*? Intuitively, for a particular state $s$, if its value has converged to the NE value, that is, $V_i(s) = V_i^*(s)$, we should no longer train on the subgame induced by it. By contrast, if the gap between its current value and the NE value is substantial, we should probably train more on the induced subgame. Thus, a simple way is to use the squared difference of the current value and the NE value as the weight for a state and sample states with probabilities proportional to the weights. Concretely, the state weight can be written as

$$w(s) = \frac{1}{2} \sum_{i=1}^{2} (V_i^*(s) - V_i(s))^2 \tag{7}$$

$$= \mathbb{E}_i\big[(V_1^*(s) - \tilde{V}_i(s))^2\big] \tag{8}$$

$$= \mathbb{E}_i\big[V_1^*(s) - \tilde{V}_i(s)\big]^2 + \text{Var}_i\big[V_1^*(s) - \tilde{V}_i(s)\big], \tag{9}$$

where $\tilde{V}_1(s) = V_1(s)$ and $\tilde{V}_2(s) = -V_2(s)$. The second equality holds because the game is zero-sum and $V_2^*(s) = -V_1^*(s)$. With random initialization and different training samples, $\{\tilde{V}_i\}_{i=1}^{2}$ can be regarded as an ensemble of two value functions and the weight $w(s)$ becomes the expectation over the ensemble. The last equality further expands the expectation to a bias term and a variance term, and we sample state with probability $P(s) = w(s)/\sum_{s'} w(s')$. For the motivating example of $RPS(n)$ game, the NE value decreases exponentially from the last state $s_{n-1}$ to the initial state $s_0$. With value functions initialized close to zero, the prioritized subgames throughout training will move gradually from the last round to the first round, which is approximately the optimal order.

However, Eq. (9) is very hard to compute in practice because the NE value is generally unknown. Inspired by Eq. (9), we propose the following alternative state weight

$$\tilde{w}(s) = \alpha \cdot \mathbb{E}_i\big[\tilde{V}_i^{(t)}(s) - \tilde{V}_i^{(t-1)}(s)\big]^2 + \text{Var}_i\big[\tilde{V}_i(s)\big], \tag{10}$$

which takes a hyperparameter $\alpha$ and uses the difference between two consecutive value function checkpoints instead of the difference between the NE value and the current value in Eq. (9). The first term in Eq. (10) measures how fast the value functions change over time. If this term is large, the value functions are changing constantly and still far from the NE value; if this term is marginal, the value functions are probably close to the converged NE value. The second term in Eq. (10) measures

**Algorithm 2:** Subgame Automatic Curriculum Learning (SACL)

**Input:** state buffers $\mathcal{M}$ with capacity $K$, probability $p$ to sample initial state from the state buffer.

Randomly initialize policy $\pi_i$ and value function $V_i$ for player $i = 1, 2$;

**repeat**

$\quad$ $V_i' \leftarrow V_i,\ i = 1, 2$;

$\quad$ // Select subgame and train policy.

$\quad$ Sample $s^0 \sim \mathrm{sampler}(\mathcal{M})$ with probability $p$, else $s^0 \sim \rho(\cdot)$;

$\quad$ Rollout in $\mathcal{MG}(s^0)$ and train $\{\pi_i, V_i\}_{i=1}^2$ via MARL;

$\quad$ // Compute weight by Eq. (10) and update state buffer.

$\quad$ $\tilde{w}^t \leftarrow \alpha \cdot \mathbb{E}[\tilde{V}_i(s^t) - \tilde{V}_i'(s^t)]^2 + \mathrm{Var}(\{\tilde{V}_i(s^t)\}_{i=1}^2),\ t = 0, \cdots, T$;

$\quad$ $\mathcal{M} \leftarrow \mathcal{M} \cup \{(s^t, \tilde{w}^t)\}_{t=0}^T$;

$\quad$ **if** $\|\mathcal{M}\| > K$ **then**

$\quad\quad$ $\mathcal{M} \leftarrow \mathrm{FPS}(\mathcal{M}, K)$;

**until** $(\pi_1, \pi_2)$ *converges*;

**Output:** final policy $(\pi_1, \pi_2)$.

the uncertainty of the current learned values and is the same as the variance term in Eq. (9) because $V_1^*(s)$ is a constant. If $\alpha = 1$, Eq. (10) approximates Eq. (9) as $t$ increases. It is also possible to train an ensemble of value functions for each player to further improve the empirical performance. Additional analysis can be found in Appendix A.3.

Since Eq. (10) does not require the unknown NE value to compute, it can be used in practice as the weight for state sampling and can be implemented for most MARL algorithms. By selecting states with fast value change and high uncertainty, our framework prioritizes subgames where agents' performance can quickly improve through learning.

### 4.3 Particle-based subgame sampler

With the sample weight at hand, we can generate subgames by sampling initial states from the state space. But it is impractical to sample from the entire space which is usually unavailable and can be exponentially large for complex games. Typical solutions include training a generative adversarial network (GAN) [11] or using a parametric Gaussian mixture model (GMM) [35] to generate states for automatic curriculum learning. However, parametric models require a large number of samples to fit accurately and cannot adapt instantly to the ever-changing weight in our case. Moreover, the distribution of weights is highly multi-modal, which is hard to capture for many generative models.

We instead adopt a particle-based approach and maintain a large state buffer $\mathcal{M}$ using all visited states throughout training to approximate the state space. Since the size of the buffer is limited while the state space can be infinitely large, it is important to keep representative samples that are sufficiently far from each other to ensure good coverage of the state space. When the number of states exceeds the buffer's capacity $K$, we use farthest point sampling (FPS) [36] which iteratively selects the farthest point from the current set of points. In our implementation, we first normalize each dimension of the state and then use the deep graph library package to utilize GPUs for fast and stable FPS results.

### 4.4 Overall algorithm

Combining the subgame sampling metric and the particle-based sampler, we present a realization of the subgame curriculum learning framework, i.e., the *Subgame Automatic Curriculum Learning* (SACL) algorithm, which is summarized in Alg. 2.

When each episode resets, we use the particle-based sampler to generate suitable initial states $s_0$ for the current policy to learn. To satisfy the requirements in Proposition 1, we also reset the game according to the initial state distribution $\rho(\cdot)$ with 0.3 probability. After collecting a number of samples, we train the policies and value functions using MARL. The weights for the newly collected states are computed according to Eq. (10) and used to update the state buffer $\mathcal{M}$. If the capacity of the state buffer is exceeded, we use FPS to select representative states-weight pairs and delete the others. An overview of SACL in the hide-and-seek game is illustrated in Fig. 3.

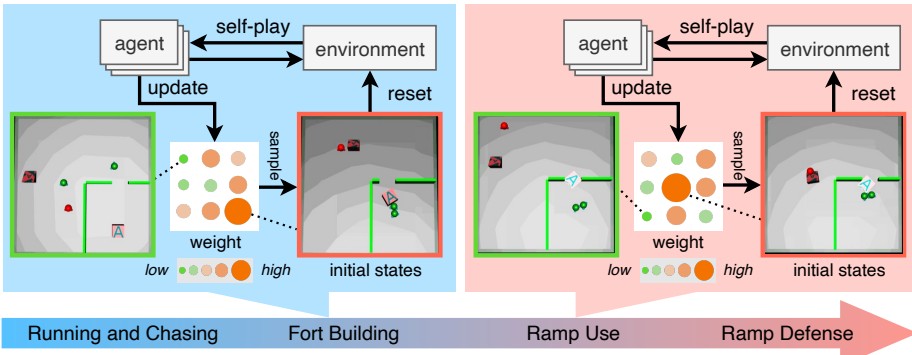

Figure 3: Illustration of SACL in the hide-and-seek environment. In the Fort Building stage, the states with hiders near the box have high weights (red) and agents can easily learn to build a fort by practicing on these subgames, while the states with randomly spawned hiders have low weights (green) and contribute less to learning. By sampling initial states with respect to the approximate squared distance to NE values, agents can proceed to new stages more efficiently.

## 5 Experiment

We evaluate SACL in three different zero-sum environments: Multi-Agent Particle Environment (MPE) [28], Google Research Football (GRF) [22], and the hide-and-seek (HnS) environment [2]. We use a state-of-the-art MARL algorithm MAPPO [52] as the backbone in all experiments.

In zero-sum games, because the performance of one player's policy depends on the other player's policy, the return curve throughout training is no longer a good evaluation method. One way to compare the performance of different policies is to use cross-play, which uses a tournament-style match between any two policies and records the results in a payoff matrix. However, due to the non-transitivity of many zero-sum games [3], winning other policies does not necessarily mean being close to NE policies, so a better way to evaluate the performance of policies is to use exploitability. Given a pair of policies $(\pi_1, \pi_2)$, the exploitability is defined as

$$\text{exploitability}(\pi_1, \pi_2) = \sum_{i=1}^{2} \max_{\pi_i'} \mathbb{E}_{s^0 \sim \rho(\cdot)} \left[ V_i^{(\pi_i', \pi_{-i})}(s^0) \right]. \tag{11}$$

Exploitability can be roughly interpreted as the "distance" to the joint NE policy. In complex environments like the ones we use, the exact exploitability cannot be calculated because we cannot traverse the policy space to find $\pi_i'$ that maximizes the value. Instead, we compute the approximate exploitability by training an approximate best response $\tilde{\pi}_i'$ of the fixed policy $\pi_i$ using MARL.

### 5.1 Main results

We first compare the performance of SACL in three environments against the following baselines for solving zero-sum games: self-play (SP), two popular variants including Fictitious Self-Play (FSP) [17] and Neural replicator dynamics (NeuRD) [19], and a population-based training method policy-space response oracles (PSRO) [23]. More implementation details can be found in Appendix B.

**Multi-Agent Particle Environment.** We consider the *predator-prey* scenario in MPE, where three slower cooperating predators chase one faster prey in a square space with two obstacles. In the default setting, all agents are spawned uniformly in the square. We also consider a harder setting where the predators are spawned in the top-right corner and the prey is spawned in the bottom-left corner. All algorithms are trained for 40M environment samples and the curves of approximate exploitability w.r.t. sample over three seeds are shown in Fig. 4(a) and 4(b). SACL converges faster and achieves lower exploitability than all baselines in both settings, and its advantage is more obvious in the hard scenario. This is because the initial state distribution in corners makes the full game challenging to solve, while SACL generates an adaptive state distribution and learns on increasingly harder subgames to accelerate NE learning. More results and discussions can be found in Appendix C.

**Google Research Football.** We evaluate SACL in three GRF academy scenarios, namely *pass and shoot*, *run pass and shoot*, and *3 vs 1 with keeper*. In all scenarios, the left team's agents cooperate

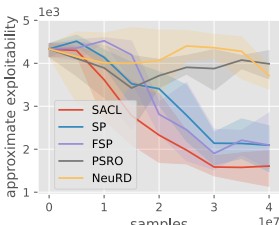
(a) MPE: exploitability.

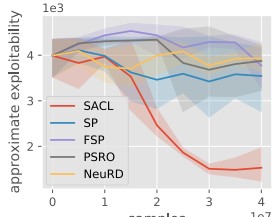
(b) MPE hard: exploitability.

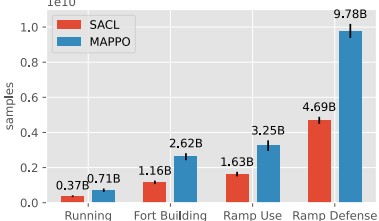
(c) HnS: number of samples.

Figure 4: Main experiment results in (a) MPE, (b) MPE hard, and (c) Hide-and-seek.

| Scenario | SACL | SP | FSP | PSRO | NeuRD |
|---|---|---|---|---|---|
| pass and shoot | **3.79 (0.87)** | 4.17 (1.45) | 4.73 (2.64) | 4.68 (2.46) | 9.18 (1.89) |
| run pass and shoot | **4.05 (1.22)** | 4.45 (1.22) | 4.62 (0.02) | 8.40 (0.48) | 9.27 (0.35) |
| 3 vs 1 with keeper | **5.49 (0.93)** | 7.76 (0.67) | 6.23 (1.14) | 7.43 (1.49) | 8.72 (0.15) |

Table 1: Approximate exploitability of learned policies in different GRF scenarios.

to score a goal and the right team's agents try to defend them. The first two scenarios are trained for 50M environment samples and the last scenario is trained for 100M samples. Table 1 lists the approximate exploitabilities of different methods' policies over three seeds, and SACL achieves the lowest exploitability. Additional cross-play results and discussions can be found in Appendix C.

**Hide-and-seek environment.** HnS is a challenging zero-sum game with known NE policies, which makes it possible for us to directly evaluate the number of samples used for NE convergence. We consider the *quadrant* scenario where there is a room with a door in the lower right corner. Two hiders, one box, and one ramp are spawned uniformly in the environment, and one seeker is spawned uniformly outside the room. Both the box and the ramp can be moved and locked by agents. The hiders aim to avoid the lines of sight from the seeker while the seeker aims to find the hiders.

There is a total of four stages of emergent stages in HnS, i.e., Running and Chasing, Fort Building, Ramp Use, and Ramp Defense. As shown in Fig. 4(c), SACL with MAPPO backbone produces all four stages and converges to the NE policy with only 50% the samples of MAPPO with self-play. We also visualize the initial state distribution to show how SACL selects appropriate subgames for agents to learn. Fig. 5(a) depicts the distribution of hiders' position in the Fort Building stage. The probabilities of states with hiders inside the room are much higher than states with hiders outside, making it easier for hiders to learn to build a fort with the box. Similarly, the distribution of the seeker's position in the Ramp Use stage is shown in Fig. 5(b), and the most sampled subgames start from states where the seeker is close to the walls and is likely to use the ramp.

## 5.2 Ablation study

We perform ablation studies to examine the effectiveness of the proposed sampling metric and particle-based sampler. All experiments are done in the hard *predator-prey* scenario of MPE and the results are averaged over three seeds. More ablation studies on state buffer size, subgame sample probability, and other hyperparameters can be found in Appendix C.

**Subgame sampling metric.** The sampling metric used in SACL follows Eq. (10) which consists of a bias term and a variance term. We compare it with four other metrics including a uniform metric, a bias-only metric, a variance-only metric, and a temporal difference (TD) error metric. The last metric uses the TD error $|\delta_t| = |r^t + \gamma V(s^{t+1}) - V(s^t)|$ as the weight, which can be regarded as an estimation of value uncertainty. The results are shown in Fig. 5(c) and the sampling metric used by SACL achieves the best results and outperforms both the bias-only metric and variance-only metric.

**State generator and buffer update method.** We substitute the particle-based sampler with other state generators including using GAN from the work [11] and using GMM from the work [35]. We also replace the FPS buffer update method with a uniform one that randomly keeps states and a greedy one that keeps states with the highest weights. Results in Fig. 5(c) show that our particle-based sampler with FPS update leads to the fastest convergence and lowest exploitability.

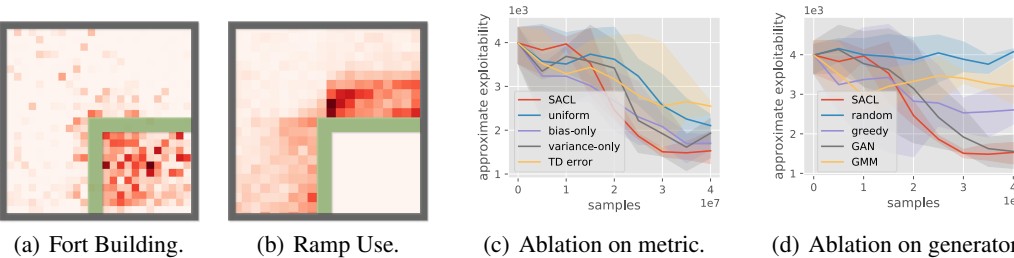

| (a) Fort Building. | (b) Ramp Use. | (c) Ablation on metric. | (d) Ablation on generator. |

Figure 5: Visualization of the state distributions in HnS (a-b) and ablation studies (c-d).

## 6 Related work

A large number of works achieve faster convergence in zero-sum games by playing against an increasingly stronger policy. The most popular methods are self-play and its variants [18, 1, 21, 34]. Self-play creates a natural curriculum and leads to emergent complex skills and behaviors [4, 2]. Population-based training like double oracle [31] and policy-space response oracles (PSRO) [23] extend self-play by training a pool of policies. Some follow-up works further accelerate training by constructing a smart mixing strategy over the policy pool according to the policy landscape [3, 33, 26, 12]. [30] extends PSRO to extensive-form games by building policy mixtures at all states rather than only the initial states, but it still directly solves the full game starting from some fixed states.

In addition to policy-level curriculum learning methods, other works to accelerate training in zero-sum games usually adopt heuristics and domain knowledge like the number of agents [27, 49] or environment specifications [5, 40, 44]. By contrast, our method automatically generates a curriculum over subgames without domain knowledge and only requires the environments can be reset to desired states. Subgame-solving technique [7] is also used in online strategy refinement to improve the blueprint strategy of a simplified abstract game. Another closely related work to our method is [9] which combines backward induction with policy learning, but this method requires knowledge of the game topology and can only be applied to finite-horizon Markov games.

Besides zero-sum games, curriculum learning is also studied in cooperative settings. The problem is often formalized as goal-conditioned RL where the agents need to reach a specific goal in each episode. Curriculum learning methods design or train a smart sampler to generate proper task configurations or goals that are most suitable for training advances w.r.t. some progression metric [10, 14, 13, 37, 29, 35, 11]. Such a metric typically relies on an explicit signal, such as the goal-reaching reward, success rates, or the expected value of the testing tasks. However, in the setting of zero-sum games, these explicit progression metrics become no longer valid since the value associated with a Nash equilibrium can be arbitrary. A possible implicit metric is value disagreement [53] used in goal-reaching tasks, which can be regarded as the variance term in our metric. By adding a bias term, our metric approximates the squared distance to NE values and gives better results in ablation studies.

Our work adopts a non-parametric subgame sampler which is fast to learn and naturally multi-modal, instead of training an expensive deep generative model like GAN [13]. Such an idea has been recently popularized in the literature. Some representative samplers are Gaussian mixture model [50], Stein variational inference [8], Gaussian process [32], or simply evolutionary computation [47, 48]. Technically, our method is also related to prioritized experience replay [38, 14, 24] with the difference that we maintain a buffer [50] to approximate the uniform distribution over the state space.

## 7 Conclusion

We present SACL, a general algorithm for accelerating MARL training in zero-sum Markov games based on the subgame curriculum learning framework. We propose to use the approximate squared distance to NE values as the sampling metric and use a particle-based sampler for subgames generation. Instead of starting from the fixed initial states, RL agents trained with SACL can practice more on subgames that are most suitable for the current policy to learn, thus boosting training efficiency. We report appealing experiment results that SACL efficiently discovers all emergent strategies in the challenging hide-and-seek environment and uses only half the samples of MAPPO with self-play. We hope SACL can be helpful to speed up prototype development and help make MARL training on complex zero-sum games more affordable to the community.

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
