## A Analysis and proofs

### A.1 Detailed analysis of the motivating example

We first show that the full state space of $RPS(n)$ can be covered within $\mathcal{O}(n)$ samples by using a state buffer and resetting games to the newly visited states. We start with an empty state buffer, and the game resets according to its initial state distribution $\rho(\cdot)$, which always resets the game to $s_0$. With a random exploration policy, the probability for the game to transit from $s_0$ to $s_1$ is $1/3$. Therefore, the number of samples required to visit state $s_1$ in expectation is $\mathbb{E}[n(s_1)] = 3$. After $s_1$ is visited, this new state will be stored in the state buffer. Since we select the newly visited states as the initial state, the game will be reset to state $s_1$ and the additional number of samples required to visit state $s_2$ in expectation is also $\mathbb{E}[n(s_2)] = 3$. In general, by starting from state $s_{k-1}$, the expected number of samples to visit state $s_k$ is $\mathbb{E}[n(s_k)] = 3$, $k = 1, 2, \cdots, n-1$. Therefore, the total number of samples required to cover the full state space is $\sum_{k=1}^{n-1} \mathbb{E}[n(s_k)] = 3(n-1)$, which is $\mathcal{O}(n)$.

Given that the state buffer has covered the entire state space, we then show that the NE Q-value of $RPS(n)$ can be learned by solving subgames with minimax-Q backward from $RSP(1)$ to $RPS(n)$. Consider using minimax-Q to solve $RPS(1)$, we can set the learning rate $\alpha = 1$ since the transition is deterministic, and the NE Q-value of a state-action pair $(s, \boldsymbol{a})$ can be learned when this pair is in the collected samples. Therefore, to learn the NE Q-values of $RPS(1)$, we have to collect all state-action pairs at least one time. With a random exploration policy, the number of samples required to cover all state-action pairs is $\sum_{i=1}^{9} 9/i = 25.46 < 26$. Therefore, the NE Q-values of $RPS(1)$ can be learned within 26 samples in expectation. Given that the NE Q-values of $RPS(k)$ are learned, the NE Q-values of $RPS(k+1)$ are only wrong at the first state, and can be learned within 26 episodes in expectation. Note that the expected episode length of $RPS(\infty)$ is 1.5, so the expected episode length of $RPS(k)$ is less than 1.5. Consider the episode used to learn the NE Q-values of the first state of $RPS(k+1)$, either $P_1$ wins and the expected episode length is less than $1 + 1.5 = 2.5$, or $P_1$ draws or loses and the episode length is 1. In both cases, the episode length is less than 2.5, so the number of samples used is less than $26 * 2.5 = 65$. Therefore, the total number of samples used to learn the NE Q-values from $RPS(1)$ to $RPS(n)$ is less than $65(n-1)$, which is $\mathcal{O}(n)$.

Since it takes $\mathcal{O}(n)$ samples to cover the entire state space and $\mathcal{O}(n)$ samples to learn the NE Q-values from $RPS(1)$ to $RPS(n)$, the total number of samples is still $\mathcal{O}(n)$.

### A.2 Proof of Proposition 1

**Proposition 1.** *If all initial states $s^0$ with $\rho(s^0) > 0$ are sampled infinitely often, and the backbone MARL algorithm is guaranteed to converge to an NE in zero-sum Markov games, then subgame curriculum learning also produces an NE of the original Markov game.*

*Proof.* When the policy trained by subgame curriculum learning converges, it is an NE of all subgames induced by the proposed states, including all initial states $s_0$ with $\rho(s_0) > 0$. Therefore, it is an NE of the original Markov game. $\qquad\square$

### A.3 Detailed analysis of the state sampling metric

We approximate the squared difference between the current value and the NE value by Eq. (10), i.e.

$$w(s) = \mathbb{E}_i\big[(V_1^*(s) - \tilde{V}_i(s))^2\big]$$
$$\approx \alpha \cdot \mathbb{E}_i\big[\tilde{V}_i^{(t)}(s) - \tilde{V}_i^{(t-1)}(s)\big]^2 + \mathrm{Var}_i\big[\tilde{V}_i(s)\big].$$

The first term in Eq. (10) uses a hyperparameter $\alpha$ and the difference between two consecutive value function checkpoints to estimate the difference between the current value and the NE value. As shown in Fig. 6, when the value function changes monotonically throughout training, the estimate can be regarded as a first-order approximation of the bias term. However, the value function of zero-sum games may oscillate up and down in different emergent stages (like in hide-and-seek) as shown in Fig. 7. In this case, the difference between two value function checkpoints is no longer an approximation of the distance to the NE value, but a first-order approximation of the difference between the current value and the next local minimal or local maximal value $V_1^{(*,k)}$, and the weight

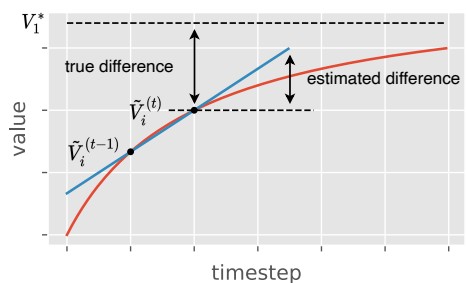

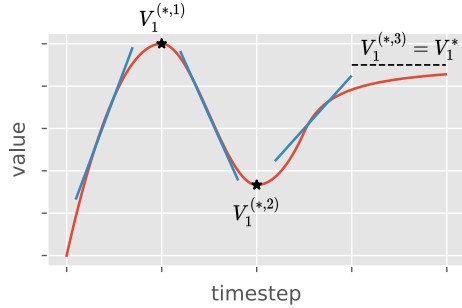

Figure 6: Approximation of the bias term when value function changes monotonically.

Figure 7: Approximation in different stages when value function oscillates in training.

becomes the approximated squared difference between the current value and the next local optimal value, i.e.,

$$
\begin{aligned}
w(s) &= \alpha \cdot \mathbb{E}_i \big[\tilde{V}_i^{(t)}(s) - \tilde{V}_i^{(t-1)}(s)\big]^2 + \mathrm{Var}_i\big[\tilde{V}_i(s)\big] \\
&\approx \mathbb{E}_i \big[V_1^{(*,k)}(s) - \tilde{V}_i(s)\big]^2 + \mathrm{Var}_i\big[V_1^{(*,k)}(s) - \tilde{V}_i(s)\big] \\
&= \mathbb{E}_i \big[(V_1^{(*,k)}(s) - \tilde{V}_i(s))^2\big].
\end{aligned}
\tag{12}
$$

Therefore, by using the weight in Eq. (10), we are not directly prioritizing states where the values are far from the NE values, but prioritizing states where the values are far from the next local optimal value. For example, in Fig. 7, before the value function has learned the first local maximal value $V_1^{(*,1)}$, we will give larger weights to states that are far from the $V_1^{(*,1)}$ to accelerate the first stage of learning $V_1^{(*,1)}$. After $V_1^{(*,1)}$ is successfully learned, we will then prioritize states that are far from the second local optimal value $V_2^{(*,1)}$ and accelerated the second stage of learning $V_1^{(*,2)}$. Finally, we learn towards the NE value $V_1^{(*,3)} = V_1^*$. By accelerating the learning in each stage, we make the NE learning process more efficient in total.

It is also possible to train an ensemble of value functions for each player to improve the estimation. Suppose we train $M$ value functions for player $i$ and denote them as $\{\tilde{V}_{i,m}\}_{m=1}^M$ for $i = 1, 2$, then the weight for state $s$ becomes

$$
w(s) = \alpha \cdot \mathbb{E}_{i,m}\big[\tilde{V}_{i,m}^{(t)}(s) - \tilde{V}_{i,m}^{(t-1)}(s)\big]^2 + \mathrm{Var}_{i,m}\big[\tilde{V}_{i,m}(s)\big],
\tag{13}
$$

where the expectation and variance are taken over both the player index $i$ and the ensemble index $m$.

## B  Implementation details

### B.1  Training details

**Multi-Agent Particle Environment.** The default and hard setting of the predator-prey scenario in MPE are shown in Fig. 8. The environment is a 2D square space and the length of a side is 4, i.e., $\{(x,y)| -2 \leq x \leq 2, -2 \leq y \leq 2\}$. 3 predators (red) cooperatively chase 1 prey (blue) and there are 2 obstacles in the space. In the default setting, all agents and obstacles are randomly spawned. In the hard setting, predators are uniformly spawned in the top-right corner, i.e., $\{(x,y)| 1 \leq x \leq 2, 1 \leq y \leq 2\}$, the prey is spawned in the bottom-left corner, i.e., $\{(x,y)| -2 \leq x \leq -1, -2 \leq y \leq -1\}$, and the obstacles are still randomly generated in the square.

This environment is fully observable, and the state of each agent is a concatenation of the positions and the velocities of all agents and the positions of all obstacles. The action space is discrete with 5 actions: idle, up, down, left, right. The environment lasts for 200 steps. In each step, if any predator collides with the prey, all predators get a reward of $+1$ and the prey gets a reward of $-1$.

The actor and critic networks use the transformer architecture. The inputs first pass through a LayerNorm layer. The normalized states are divided into different entities including self, other agents, obstacles, and time, then each entity passes through fully connected layers to get its embedding. The

| Hyperparameters | Value |
| --- | --- |
| Learning rate | 5e-4 |
| Discount rate ($\gamma$) | 0.99 |
| GAE parameter ($\lambda_{\text{GAE}}$) | 0.95 |
| Gradient clipping | 10.0 |
| Adam stepsize | 1e-5 |
| Value loss coefficient | 1 |
| Entropy coefficient | 0.01 |
| Parallel threads | 100 |
| PPO clipping | 0.2 |
| PPO epochs | 5 |
| Size of embedding layer | 32 |
| Size of MLP layer | 64 |
| Size of LSTM layer | 64 |
| Residual attention layer | 8 |
| probability $p$ | 0.7 |
| Ensemble size $M$ | 3 |
| Capacity $K$ | 10000 |
| Weight of the value difference $\alpha$ | 0.7 |

Table 2: Hyperparameters of MPE.

| Hyperparameters | Value |
| --- | --- |
| Learning rate | 5e-4 |
| Discount rate ($\gamma$) | 0.99 |
| GAE parameter ($\lambda_{\text{GAE}}$) | 0.95 |
| Gradient clipping | 10.0 |
| Adam stepsize | 1e-5 |
| Value loss coefficient | 1 |
| Entropy coefficient | 0.01 |
| Parallel threads | 200 |
| PPO clipping | 0.2 |
| PPO epochs | 10 |
| Size of MLP layer | 64 |
| probability $p$ | 0.7 |
| Ensemble size $M$ | 3 |
| Capacity $K$ | 10000 |
| Weight of the value difference $\alpha$ | 0.7 |

Table 3: Hyperparameters of GRF.

| Length | Information |
| --- | --- |
| 22 | (x,y) coordinates of left team players |
| 22 | (x,y) direction of left team players |
| 22 | (x,y) coordinates of right team players |
| 22 | (x,y) direction of right team players |
| 3 | (x, y and z) ball position |
| 3 | ball direction |
| 3 | one hot encoding of ball ownership (none, left, right) |
| 11 | one hot encoding of which player is active |
| 7 | one hot encoding of game mode |

Table 4: Information in the state vector of GRF.

weights of the embedding layers are shared within entities of the same type. Then the embedding of each entity is concatenated with the self states and passed through a self-attention network. Then we average the output of the attention block and concatenate it with the self-embedding to get the final representation. This representation is then passed through a LayerNorm layer and an MLP layer and then produces the value through a critic head and the action through an actor head. All hyperparameters for training are listed in Table 2.

**Google Research Football.** The environment is a physics-based 3D football simulation and the length and width are 2.0 and 0.9, i.e., $\{(x, y)| - 1.0 \leq x \leq 1.0, -0.45 \leq y \leq 0.45\}$. The *pass and shoot* scenario in GRF is shown in Fig. 9. There are five players and a soccer ball in the environment, with a scripted goalkeeper and two RL attackers on the left side and a scripted goalkeeper and one RL defender on the right side. The left goalkeeper is spawned at $(-1.0, 0.0)$ and the two attackers are spawned at $(0.7, 0.0)$ and $(0.7, -0.3)$. The right goalkeeper is spawned at $(1.0, 0.0)$ and the defender is spawned at $(0.75, -0.3)$. The ball is spawned at $(0.7, -0.28)$. The *run, pass and shoot* scenario in GRF is shown in Fig. 10. There are five players and a soccer ball in the environment, with a scripted goalkeeper and two RL attackers on the left side and a scripted goalkeeper and one RL defender on the right side. The left goalkeeper is spawned at $(-1.0, 0.0)$ and the two attackers are spawned at $(0.7, 0.0)$ and $(0.7, -0.3)$. The right goalkeeper is spawned at $(1.0, 0.0)$ and the defender is spawned at $(0.75, -0.1)$. The ball is spawned at $(0.7, -0.28)$. The *3 vs 1 with keeper* scenario in GRF is shown in Fig. 11. There are six players and a soccer ball in the environment, with a scripted goalkeeper and three RL attackers on the left side and a scripted goalkeeper and one RL defender on

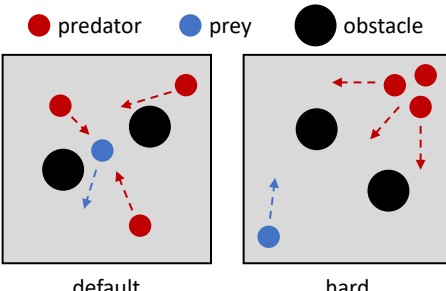

Figure 8: Illustration of the default and hard setting of predator-prey in MPE.

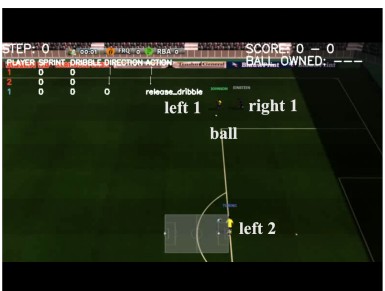

Figure 9: Pass and shoot scenario in GRF.

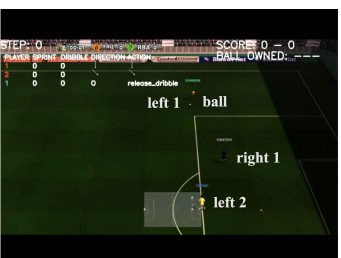

Figure 10: Run, pass and shoot scenario in GRF.

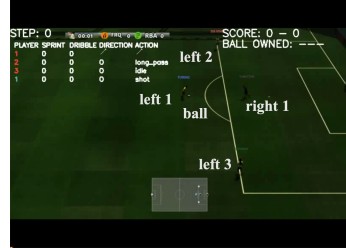

Figure 11: 3 vs 1 with keeper scenario in GRF.

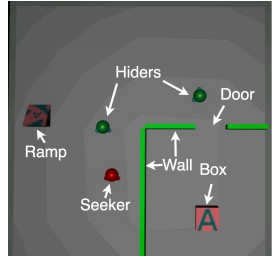

Figure 12: Quadrant scenario in HnS.

the right side. The left goalkeeper is spawned at $(-1.0, 0.0)$ and the three attackers are spawned at $(0.6, 0.0)$, $(0.7, 0.2)$ and $(0.7, -0.2)$. The right goalkeeper is spawned at $(1.0, 0.0)$ and the defender is spawned at $(0.75, 0.0)$. The ball is spawned at $(0.6, 0.0)$. In all three environments, attackers have to learn how to dribble the ball, cooperate with teammates to pass the ball, and overcome the defender's defense to score goals.

The environment is fully observable, and the state of each agent is a 115-dimensional vector, including the coordinates of left team players, the directions of left team players, the coordinates of right team players, the directions of right team players, the ball position, the ball direction, one hot encoding of ball ownership, one hot encoding of which player is active and one hot encoding of game mode. The detailed information is listed in Table 4. The action space is discrete with 19 actions: idle, left, top left, top, top right, right, bottom right, bottom, bottom left, long pass, high pass, short pass, shoot, start sprinting, reset current movement direction, stop sprinting, slide, start dribbling and stop dribbling. An episode lasts a maximum of 200 steps. The environment ends prematurely when one side scores, the possession of the ball changes, or the game is out of play. We use the standard scoring and checkpoint rewards provided by the football engine. Specifically, if the left team scores a goal in each step, all left players get a reward of +1 and the right player gets -1. There are also 10 concentric circles with the goal in the center, called checkpoint regions. The left team obtains an additional checkpoint reward of +0.1 when they possess the ball and first move into the next checkpoint region, and the right team gets -0.1. Checkpoint rewards are only given once per episode.

The inputs of the actor and critic networks first pass through a LayerNorm layer. The normalized states then pass through an MLP layer and then produce the value through a critic head and the action through an actor head. All hyperparameters for training are listed in Table 3.

**Hide-and-seek environment.** The quadrant scenario in the hide-and-seek environment is shown in Fig. 12. The environment is a square space with a square room with a door in the bottom-right corner. There are 2 hiders (green), 1 seeker (red), 1 box, and 1 ramp. At the beginning of each episode, the hiders, box, and ramp are uniformly spawned inside the room, and the seeker is uniformly spawned outside the room.

The environment is fully observable and the state of each agent is a concatenation of the positions and velocities of all agents, the positions, velocities, and lock flags of the box and the ramp, and the

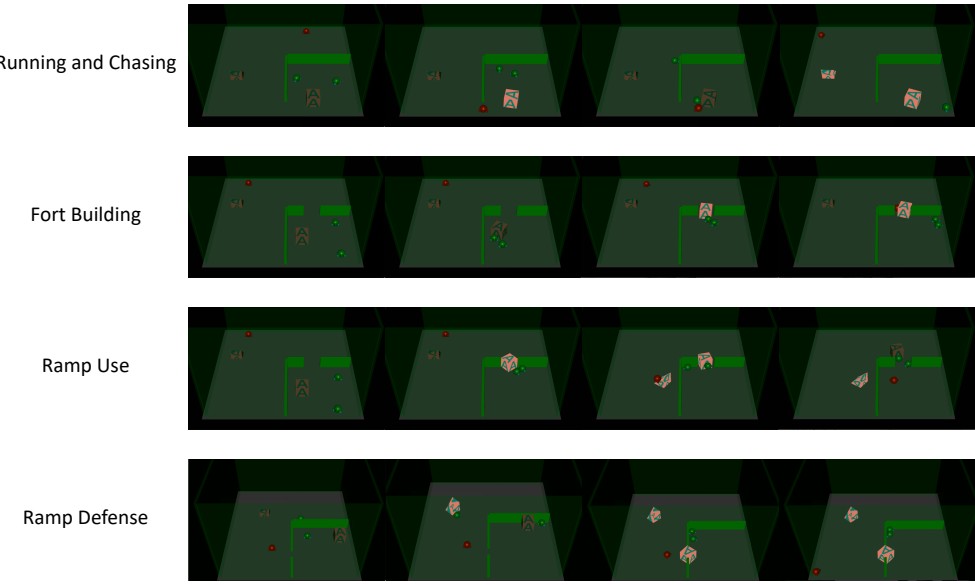

Figure 13: Sample trajectory traces from each emergent stage in quadrant scenario of HnS.

current timestep. The action space is discrete and agents can choose to move in 4 directions, grab, and lock/unlock. Each episode lasts for 80 steps and is divided into 2 phases: the preparation phase and the main phase. In the preparation phase, the seeker is fixed and only the hiders can act to prepare for the main phase. No reward is given to any agent in the preparation phase. In the main phase, all agents can act and the seeker tries to find the hiders and the hiders try to avoid being discovered. When the hiders are spotted by the seeker, the seeker gets a reward of $+1$ at this step and the hiders get a reward of $-1$. Otherwise, the seeker gets a reward of $-1$ and the hiders get $+1$.

There are a total of 4 emergent stages in this game, as shown in Fig. 13. (1) *Running and Chasing*: The hiders learn to run away from the seeker to avoid detection, while the seeker learns to chase the hiders. The seeker is the winner at this stage and the average episode reward of hiders is about $-20$. (2) *Fort Building*: In the preparation phase, the hiders learn to use the box to block the door and lock it in place to build a fort so that the seeker cannot enter the room and see the hider. The hiders are the winner in this stage, and the average episode reward of hiders is about 30. (3) *Ramp Use*: The seeker learns to move the ramp to the wall of the room and use it to get into the room. The average episode reward of hiders reduces to about 25 but is still larger than 0. (4) *Ramp Defense*: In the preparation phase, the hiders learn to move the ramp into the room or push it far away from the wall and lock it to prevent being used by the seeker. The seeker can no longer enter the room and find the hiders. The average episode reward of hiders is about 40 at this stage.

We adopt the same network architecture as [2]. The states are divided into different entities including self, other agents, box, and ramp, then each entity passes through fully connected layers to get its embedding. The weights of the embedding layers are shared within entities of the same type. Then the embedding of each entity is concatenated with the self embedding and passed through a self-attention network. Then we average the output of the attention block and concatenate it with the self-embedding to get the final representation. This representation is then passed through an MLP layer and a LSTM layer and then produces the value through a critic head and the action through an actor head. All hyperparameters of HnS are listed in Table 5.

Besides zero-sum games, it is also possible to use SACL in cooperative tasks. We choose the Ramp Use stage in HnS to show that SACL can produce comparable results to curriculum learning algorithms specialized for cooperative tasks [8]. In this task, there is 1 hider with fixed policy, 1 seeker to train, 1 box and 1 ramp. We need to train a seeker policy to use the ramp to get into the quadrant room for positive rewards. The environment is fully observable and the state is the same as that in the quadrant scenario. We use the same prior knowledge to define easy tasks as [8],

| Hyper-parameters | Value |
|---|---|
| Learning rate | 3e-4 |
| Discount rate ($\gamma$) | 0.998 |
| GAE parameter ($\lambda_{\text{GAE}}$) | 0.95 |
| Gradient clipping | 5.0 |
| Adam stepsize | 1e-5 |
| Value loss coefficient | 1 |
| Entropy coefficient | 0.01 |
| PPO clipping | 0.2 |
| Chunk length | 10 |
| PPO epochs | 4 |
| Horizon | 80 |
| Mini-batch size | 64000 |
| Size of embedding layer | 128 |
| Size of MLP layer | 256 |
| Size of LSTM layer | 256 |
| Residual attention layer | 32 |
| Weight decay coefficient | $10^{-6}$ |
| probability $p$ | 0.7 |
| Ensemble size $M$ | 3 |
| Capacity $K$ | 10000 |
| Weight of the value difference $\alpha$ | 1.0 |

Table 5: Hyperparameters of HnS.

| Hyperparameters | Value |
|---|---|
| Learning rate | 5e-4 |
| Discount rate ($\gamma$) | 0.99 |
| GAE parameter ($\lambda_{\text{GAE}}$) | 0.95 |
| Gradient clipping | 20.0 |
| Adam stepsize | 1e-5 |
| Value loss coefficient | 1 |
| Entropy coefficient | 0.01 |
| PPO clipping | 0.2 |
| chunk length | 10 |
| PPO epochs | 15 |
| Horizon | 60 |
| Parallel threads | 300 |
| probability $p$ | 0.7 |
| Ensemble size $M$ | 3 |
| Capacity $K$ | 2000 |

Table 6: Hyperparameters of the cooperative task in HnS

which prioritizes states where the ramp is right next to the wall and agents are next to the ramp. All hyperparameters are listed in Table 6.

## B.2 Evaluation details

**Exploitability.** We compute the approximate exploitability by training an approximate best response $\tilde{\pi}'_i$ of the fixed policy $\pi_i$ using MAPPO. The lower the exploitability, the better the algorithm. We use the checkpoints of an algorithm's policy trained with different numbers of environment steps to estimate the exploitability. Specifically, we run SACL in MPE and save a policy checkpoint when the agent has consumed 0M, 5M, 10M, 15M, ..., and 40M environment samples. Then for each checkpoint, we keep it fixed and train an adversarial policy to be the best response of the fixed policy to estimate the exploitability. Then we get an exploitability curve of SACL over samples. Finally, we repeat this procedure for two more seeds and average the results and plot the std error. For a single algorithm, we trained $9 \times 3 = 27$ (checkpoints $\times$ seeds) best-response policies to plot one curve in the exploitability graph.

**Cross-play.** We evaluate SACL and other baselines by cross-play, which uses a head-to-head match between any two policies and records the results in a payoff matrix. In MPE, the element of the payoff matrix represents the episodic reward of the predators, and in GRF, represents the win rate of the red team. More specifically, we train 3 seeds for each algorithm and match three models of one algorithm against the three models of the opponent algorithm, i.e., we get $3 \times 3 = 9$ competitions between any two algorithms and report the average results and the std error. For example, in MPE, we use three different predators of SACL to compete with three different preys of SP to get the episode predator reward. We can evaluate the performance of the predator using the elements of a row and evaluate the performance of the prey using the elements of a column. We use the first row to represent the predator of SACL, then a larger value in this row than other rows means that the predator of SACL is better than other algorithms. We use the first column to represent the prey of SACL, then a smaller value in this column than other columns means that the prey of SACL is better than other algorithms.

**Four rounds of emergent strategies in HnS.** As shown in Figure 14, we use three inflection points to evaluate the sample required to produce the first three stages. More specifically, the *Running and Chasing* phase ends when the hider's reward decreases to the lowest value of about $-20$. When the hider's reward begins to increase, the *Fort-Building* phase begins and continues until the hider's

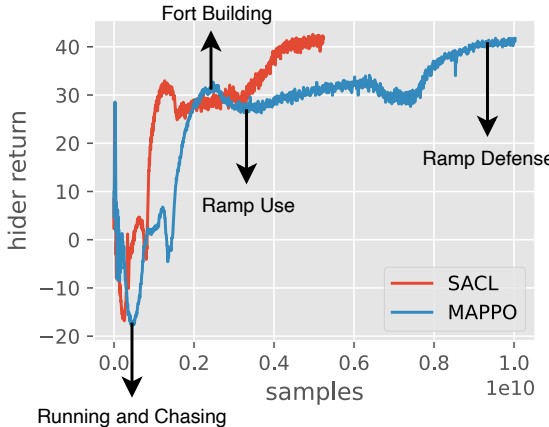

Figure 14: Checkpoints of four rounds of emergent strategies in HnS.

reward reaches a local maximum of about 30. Then the agents move to the *Ramp-Defense* phase until the hider's reward reaches a local minimum and begins the final *Ramp-Use* stage. We choose the point when the hider's episode reward reaches 40 as the end of the final stage.

## C   Additional experiment results

### C.1   Multi-Agent Particle Environment

**Cross-play.** The results of cross-play at $40M$ in MPE and MPE hard are shown in Fig. 15 and Fig. 16. In MPE and MPE hard, the predator and prey of SACL beat all baselines. For example, let x be the row x and y be represent the column y of the payoff matrix. We compare the predator of SACL with FSP using rows 1 and 3 and find that the elements of row 1 are larger than the elements of row 3, i.e., the predator of SACL is better than FSP. The elements of column 1 are smaller than the elements of column 3, which means the prey of SACL is better than FSP. The prey trained by SACL swerves to avoid the predators when the predators surround him and the predators learn to capture the prey in the two environments. SP is comparable with SACL in MPE, but in the hard setting, SP does not converge to the NE policy due to the large initial distance between predator and prey. We show the initial state distributions of the predator in SP and SACL at $40M$ training steps in Fig. 17. We find that in MPE hard, the initial distance between the prey and the predator is too far. As a result, the prey trained by SP learns little about how to stay away from predators and the predators have hardly learned how to catch the prey. FSP also performs worse than SACL in the hard setting for the same reason as SP. For PSRO, it is even difficult to obtain the best response corresponding to the prey of random policy in MPE hard because the initial distance between prey and predator is too far. NeuRD performs poorly in both environments because NeuRD's update rules cause drastic policy changes and erratic convergence.

### C.2   Google Research Football

The results of cross-play in *pass and shoot*, *run, pass and shoot* and *3 vs 1 with keeper* are shown in Fig. 18. In the three scenarios, SACL is comparable to FSP and PSRO, and better than SP and NeuRD. For example, let x be the row x and y be the column y of the payoff matrix. In *3 vs 1 with keeper*, the elements of row 1 are larger than the elements of row 2, which means the attackers of SACL are better than SP. The elements of column 1 are comparable with the elements of column 2, i.e., the prey of SACL is comparable with SP. It is worth mentioning that in *run, pass and shoot*, FSP and PSRO attackers have a higher win rate than SACL against PSRO and NeuRD defenders. This is because PSRO and NeuRD defenders have a bad defensive policy, and FSP and PSRO attackers have their counter policy. However, Table 1 in the main text shows that the exploitability of SACL is lower than others. This is because zero-sum games are non-transitive. For example, in rock-paper-scissors, it doesn't mean that rock is better than paper just because rock beats scissors and scissors beats

Predator Reward

| | SACL | SP | FSP | PSRO | NeuRD |
|---|---|---|---|---|---|
| SACL | 59.69 (41.72) | 61.21 (25.92) | 162.09 (55.26) | 1464.72 (695.34) | 789.59 (301.62) |
| SP | 58.87 (25.86) | 53.21 (21.61) | 148.80 (40.10) | 1095.97 (564.00) | 680.49 (249.03) |
| FSP | 26.31 (31.11) | 11.37 (9.71) | 43.13 (36.33) | 1589.40 (267.55) | 1679.08 (630.54) |
| PSRO | 16.99 (41.64) | 3.94 (4.62) | 6.32 (8.10) | 439.40 (626.79) | 1332.36 (959.75) |
| NeuRD | 0.47 (0.44) | 0.41 (0.27) | 0.38 (0.40) | 5.13 (13.16) | 154.98 (71.51) |

Figure 15: MPE: cross-play

Predator Reward

| | SACL | SP | FSP | PSRO | NeuRD |
|---|---|---|---|---|---|
| SACL | 34.33 (10.38) | 596.68 (431.65) | 445.01 (416.05) | 868.84 (421.43) | 970.54 (468.76) |
| SP | 0.09 (0.25) | 14.93 (39.14) | 63.71 (171.49) | 13.08 (33.88) | 114.12 (165.99) |
| FSP | 16.71 (22.76) | 859.64 (838.72) | 273.86 (543.29) | 634.86 (623.71) | 1338.70 (1226.54) |
| PSRO | 4.14 (8.62) | 304.96 (631.03) | 30.44 (32.28) | 134.93 (130.23) | 997.39 (1390.75) |
| NeuRD | 0.02 (0.06) | 3.96 (8.50) | 2.87 (4.79) | 19.53 (36.64) | 25.51 (28.25) |

Figure 16: MPE hard: cross-play.

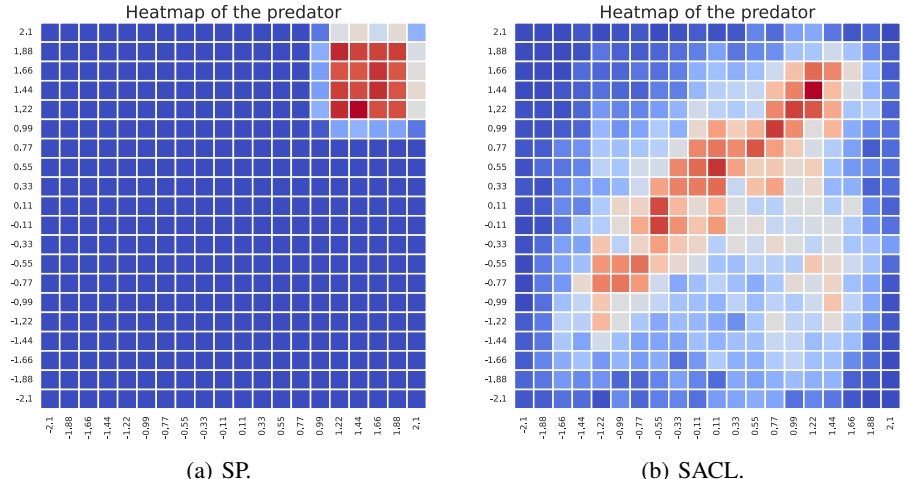

(a) SP.

(b) SACL.

Figure 17: Visualization of the state distributions in MPE hard.

paper. Thus, a high return against a single policy does not mean that it is close to the NE policy, and the comparable result in cross-play does not contradict with the exploitability result. In general, exploitability is a better measure of policy performance and is used in many papers.

We also visualize the behavior of different methods to show that SACL learns more complex policies than others and is closer to the NE policies. For example, in *3 vs 1 with keeper*, the NE policy is that the left players shoot from the top, middle, and bottom with equal probability. SACL learns to shoot from the top and the middle, while FSP and PSRO only shoot from the bottom.

### C.3 Hide-and-seek

Although SACL is derived for zero-sum games, it is also applicable to more general settings such as goal-conditioned problems. We consider the *Ramp-Use* task proposed in VACL [8], where the seeker aims to get into the lower-right quadrant (with no door opening) which is only possible by using a ramp. We adopt the same prior knowledge of "easy tasks" used in VACL to initialize the state buffer $\mathcal{M}$ and achieve comparable sample efficiency with VACL, one of the strongest ACL algorithms for goal-conditioned RL. The result is shown in Figure 19.

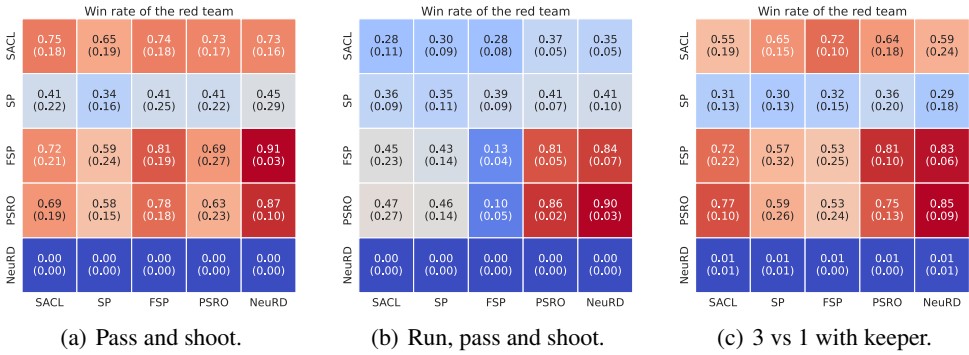

(a) Pass and shoot.   (b) Run, pass and shoot.   (c) 3 vs 1 with keeper.

Figure 18: The results of cross-play in GRF.

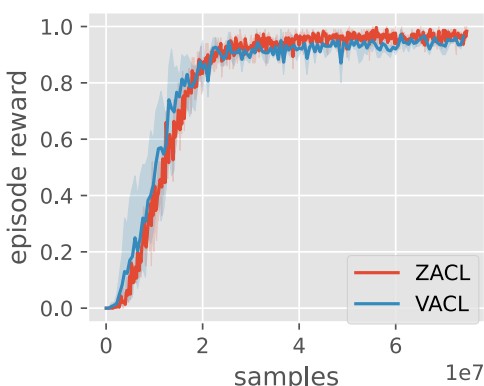

Figure 19: Seeker's average episode rewards in a goal-reaching *Ramp-Use* task. SACL is comparable to a strong baseline VACL, which is specialized for goal-oriented problems.

## C.4 Ablation studies

**Buffer size.** As shown in Fig. 20(a), the buffer capacity $K$ must be large enough. When the buffer is too small, the states in the buffer cannot approximate the state space. When the buffer is too large, FPS consumes much time. So we finally choose $K = 10000$.

**Subgame sample probability.** As shown in Fig. 20(b), we need more samples from the subgame buffer than uniform sampling in the training batch, and uniform sampling from the state space ensures global exploration. When $p$ is too small, SACL degenerates into SP, resulting in poor performance. When $p = 1$, the lack of global exploration also leads to poor performance. Finally we choose $p = 0.7$.

**Ensemble size.** As shown in Fig. 20(c), we can train an ensemble of value functions for each player to improve the estimation. Excessive ensemble size requires much memory and training time. So we finally choose $M = 3$.

**Weight of the value difference.** As shown in Fig. 20(d), our algorithm is insensitive to the Weight of the value difference $\alpha$. Empirically, we prefer $\alpha$ less than 1. We finally choose $\alpha = 0.7$ in MPE, MPE hard and GRF, $\alpha = 1.0$ in Hns.

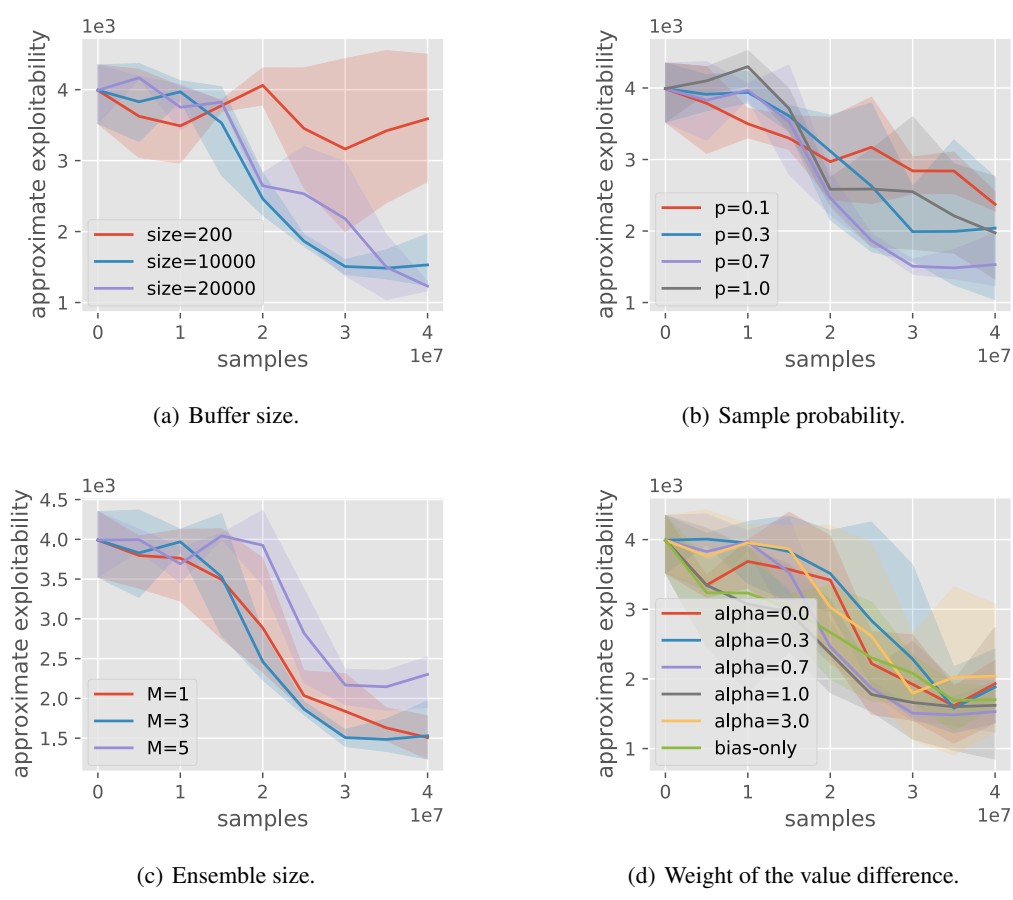

(a) Buffer size.

(b) Sample probability.

(c) Ensemble size.

(d) Weight of the value difference.

Figure 20: Ablation studies of hyperparameters in MPE hard.