# OpenReview forum: "Accelerate Multi-Agent Reinforcement Learning in Zero-Sum Games with Subgame Curriculum Learning"
_NeurIPS.cc/2023/Conference — Submitted to NeurIPS 2023_

### Official Review · Reviewer_aeUw · 2023-07-04

**Soundness:** 3 good
**Presentation:** 3 good
**Contribution:** 2 fair
**Rating:** 5
**Confidence:** 3

**Summary:**

The paper proposes a general subgame curriculum learning framework to accelerate MARL training for zero-sum games. It adopts an adaptive initial state distribution by resetting agents to some previously visited states where they can quickly learn to improve performance. The author derives a subgame selection metric that approximates the squared distance to NE values and further adopt a particle-based state sampler for subgame generation.

**Strengths:**

The paper presents a general framework to accelerate NE learning in zero-sum games by training over a curriculum of subgames. The author develops an automatic curriculum learning algorithm, i.e., Subgame Automatic Curriculum Learning, which can adopt any MARL algorithm as its backbone and preserve the overall convergence property. The paper is well written. A motivating example is also described to illustrate the main idea, which is beneficial for readers to understand the work.


**Weaknesses:**

1. As we can see, the convergent speed is accelerated by the proposed method. However, there is no evidence that the final performance is improved.
2. The core part of this work is to compute the weight of the state. It would be better to compare with other metrics.



**Questions:**

I mainly have the following concerns.
1.How about the performance improvement brought by the proposed method?
2.Would it be better if other metrics, for example, the least visited times of a state, are used solely or combined with the proposed metric in Section4.2? It would be better the see more comparison results about this.
3.It seems there are mistakes in Eq. (8) and Eq. (9). The index 2 is missing from (7) to (8). How the equation is transformed from (8) to (9).


**Limitations:**

The author could make a discussion about the application of the proposed method on three or more players.

---

> ### Author Rebuttal · Authors · 2023-08-09
>
> Thank you for your time and valuable comments! We appreciate your acknowledgment of our proposed framework and recognition of the illustrative example. We hope our responses can address your concerns.
>
> **Q1: The convergent speed is accelerated by the proposed method. How about the final performance improvement?**
>
> In zero-sum games, the optimal solution is the Nash equilibrium (NE) strategy. And according to the Minimax theorem [1], the value of the NE is unique in zero-sum games. Therefore, in theory, we cannot change the final performance of NE strategies in zero-sum games.
>
> In practice, it is very hard to learn the exact NE strategies in complex games like GRF. So the performance of different algorithms is often evaluated by using the same amount of training samples and checking the policy's exploitability. In this case, faster convergence means better performance and our results in MPE and GRF show that SACL achieves the lowest exploitability in all algorithms.
>
> [1] Von Neumann, John, and Oskar Morgenstern. "Theory of games and economic behavior, 2nd rev." (1947).
>
> **Q2: It would be better if other metrics, for example, the least visited times of a state, are used solely or combined with the proposed metric in Section 4.2.**
>
> Thank you for your suggestion, we added the least visited metric to compare it with our proposed metric, and the result is shown in Fig. 3(a) of the global response PDF. The four metrics in our ablation study (Section 5.2) is included for comparison: uniform, bias-only, variance-only, and TD error. We also consider combining our metric $w_{\text{SACL}}(\cdot)$ with the least visited metric $w_{\text{lv}}(\cdot)$, i.e., $w(s) = w_{\text{SACL}}(s) + \lambda\cdot w_{\text{lv}}(s)$, and the result is shown in Fig. 3(b) of the global response PDF.
>
> As shown in the results, our proposed metric has the best performance and combining with the least visited metric does not improve the result. The least visited metric does not work well in zero-sum games because the importance of a state depends not only on the visited times, but also on the opponents' policy. Even if a state has been visited many times, if the opponents' policy has changed, the agents still need to be trained on the induced subgame. Therefore, it is not a suitable subgame sampling metric in zero-sum games.
>
>
> **Q3: It seems there are mistakes in Eq. (8) and Eq. (9). The index 2 is missing from (7) to (8). How the equation is transformed from (8) to (9).**
>
> The derivation of Eq. (8) and Eq. (9) omitted a few steps, so it may be a little confusing but there is no mistake.
> * From Eq. (7) to Eq. (8), we use the fact that $V_2^*(s) = -V_1^*(s)$ (zero-sum) and $V_1(s)=\tilde{V_1}(s)$, $V_2(s)=-\tilde{V_2}(s)$ (definition), and the detailed derivation is:
> $$
> \begin{aligned}
> w(s) &= \frac{1}{2} \sum_{i=1}^2 (V_i^*(s) - V_i(s))^2 \newline
> &= \frac{1}{2} (V_1^*(s) - \tilde{V}_1(s))^2 + \frac{1}{2} (-V_1^*(s) + \tilde{V}_2(s))^2 \newline
> &=\mathbb{E}_i[(V_1^*(s) - \tilde{V}_i(s))^2].
> \end{aligned}
> $$
> * From Eq. (8) to Eq. (9), we use the fact that $\mathbb{E}[A^2] = \mathbb{E}[A]^2 + \text{Var}[A]$.
>
> ---
>
> We genuinely value your dedication to reviewing our paper and believe our detailed responses have addressed your concerns. We would really appreciate it if you could consider raising the rating of our work based on our responses.

---

> > ### Comment · Reviewer_aeUw · 2023-08-18
> > **Thanks for your explanation.**
> >
> > Thanks for your explanation.

---

### Official Review · Reviewer_9JJp · 2023-07-04

**Soundness:** 3 good
**Presentation:** 4 excellent
**Contribution:** 3 good
**Rating:** 5
**Confidence:** 4

**Summary:**

The paper proposes a novel subgame curriculum learning framework for accelerating multi-agent reinforcement learning (MARL) in zero-sum Markov games. The framework uses an adaptive initial state distribution to induce subgames of varying difficulty for agents to learn, and leverages a sampling metric that approximates the squared distance to Nash equilibrium (NE) values to prioritize subgames with fast value change and high uncertainty. The paper instantiates the framework with a particle-based state sampler and integrates it with any MARL algorithm, resulting in the Subgame Automatic Curriculum Learning (SACL) algorithm. The paper evaluates SACL in three zero-sum environments and show that it converges faster and achieves lower exploitability than existing methods.

**Strengths:**


- The paper addresses an important and challenging problem of reducing the computational cost of solving complex zero-sum games with MARL, which has many potential applications and implications.
- The paper provides a illustrative motivating example to justify the effectiveness of the subgame curriculum learning framework, which leverages ideas from goal-conditioned RL and prioritized experience replay.
- The paper is able to provide practical methods to approximately solve the hard parts of the theoretical work. For example, the paper introduces a novel subgame sampling metric that approximates the squared distance to NE values with a bias term and a variance term. The paper also adopts a particle-based state sampler that is compatible with most MARL algorithms.
- The paper conducts extensive experiments on three different zero-sum environments and demonstrates that SACL can converge to NE policies with substantially fewer samples. The ablation study part is able to demonstrate how well the approximation methods in SACL work comparing to other alternatives.

**Weaknesses:**


- The paper assumes that the environment can be reset to any desired state to generate induced subgames, which may not be feasible or realistic in some settings.
- In Section 5.1, it is not clear how the approximate exploitability is computed using MARL methods for best response training. What are the exact algorithms used for this purpose? How many samples are used for each best response training? How reliable are these estimates? Some details on these aspects would help evaluate the results more fairly.
- In Section 5.2, it would be helpful to report some quantitative results on how different choices of state buffer size, hyperparameters alpha, subgame sample probability, or other hyperparameters affect SACL's performance or convergence.
- It would also be helpful to have more details on how to implement FPS for buffer update and how to measure the distance between states.

**Questions:**


- In the subgame sampling metric, the difference between two consecutive value function checkpoints is used to approximate the difference between the NE value and the current value. What if the value function goes into a bad local minimum and stops moving? Has this been observed in any experiment?
- In Section 5.2, the paper reports that the subgame sampling metric works better than the TD error in practice. Since the value function is often updated based on TD error between each iteration, what is the intuition of the subgame sampling metric being better?
- How does SACL handle cases where there are multiple NEs in a game? Does it converge to one randomly?
- How generalizable is SACL to other types of games such as cooperative games or general-sum games? What modifications or extensions are needed to adapt SACL to these settings?

**Limitations:**


- The paper requires access to full state information and reset function for each environment.
- The paper relies on approximate exploitability as a proxy for measuring closeness to NE policies. However, approximate exploitability may not reflect the true performance gap between different policies due to sample variance or approximation errors.

---

> ### Author Rebuttal · Authors · 2023-08-09
>
> Thank you for your appreciation of our work and thoughtful comments! We are encouraged to see your acknowledgment of our work’s novelty and the positive assessment of our experiment results. We hope our responses can address your concerns.
>
> **Q1: The assumption that environments can be reset to any state may not be feasible or realistic in some settings.**
>
> We agree that some environments don't natively support this assumption. But this feature is easy to implement by modifying the environment’s reset function. In fact, the MPE environment in our experiment doesn’t satisfy this requirement, but a little change to its `reset_world()` function makes it works with our framework.
>
> We would also like to remark that this assumption is very common in the curriculum learning literature and is widely used in works like [1, 2], where setting tasks is equivalent to resetting initial states in our case.
>
> [1] Florensa, Carlos, et al. "Automatic goal generation for reinforcement learning agents." ICML, 2018.
>
> [3] Portelas, Rémy, et al. "Teacher algorithms for curriculum learning of deep rl in continuously parameterized environments." CoRL, 2020.
>
> **Q2: How is the approximate exploitability computed using MARL methods?**
>
> The implementation details are included in Appendix B.2 of the supplementary material. We will add more information to the main text in the next version.
>
> We use MAPPO to train an approximate best response (BR). In principle, the BR is trained until convergence. A BR is trained for 200M samples in MPE and 400M in GRF. We ensure our results are reliable in the following ways:
> 1. we use much more samples to train the BR than we used to train NE policies (200M >> 40M, 400M > 50M/100M) to ensure the BRs are fully converged.
> 2. We train individual BR for different seeds and report the mean and std. A single curve in the exploitability figures requires $9 \times 3 = 27$ (checkpoint $\times$ seeds) trained BRs.
> 3. We render the behaviors of different algorithms' policies to validate that SACL does learn a stronger strategy. The gifs can be found on our [website](https://sites.google.com/view/sacl-neurips).
>
> **Q3: More quantitative results on how different hyperparameters affect SACL's performance.**
>
> All these ablation studies are included in Appendix C.4 and shown in Fig. 20. Please see the detailed discussion in the appendix.
>
> **Q4: More details on how to implement FPS for buffer updates and how to measure the distance between states.**
>
> In general, FPS iteratively selects the farthest point from the current set of points. The distance between two states is simply the Euclidean distance. The distance between a state $s_a$ and a set of states $S$ is the smallest distance between $s_a$ and any state in $S$, i.e., $\text{min}_{s\in S} \|s_a - s\|$. For implementation, we first normalize each dimension of the state vector to make all values lie in the range $[0, 1]$. Then we directly use the `farthest_point_sampler()` function from the [Deep Graph Library](https://docs.dgl.ai/en/latest/api/python/dgl.geometry.html#farthest-point-sampler) to utilize GPUs for fast and stable results.
>
> **Q5: In the subgame sampling metric, what if the value function goes into a bad local minimum and stops moving?**
>
> If we only consider the bias term in the metric, falling into a local minimum would indeed make the metric fail. Fortunately, we also include a variance term of several value functions initialized with different seeds and trained with different samples. If one value function goes into a bad local minimum, it will make the variance large, and further make the weight large so that the policies are trained more on this unsolved subgame.
>
> **Q6: What is the intuition of the subgame sampling metric being better than the TD error metric?**
>
> One reason is that the TD error metric only measures the change of value functions, which is similar to the bias term in our metric. We have an additional variance term to consider the uncertainty and thus get better performance. Another reason is that TD error is less stable because it always uses the latest value. While the bias term in our metric is computed using two consecutive checkpoints, which are usually 5-10 iterations apart and more stable.
>
> **Q7: How does SACL handle cases where there are multiple NEs in a game?**
>
> In zero-sum games, although there may be multiple NE strategies, their NE values are the same [4]. This means that any NE strategy is optimal. Therefore, we only aim to learn an NE but don't consider which NE. In practice, since SACL reserves the convergence property of the backbone MARL algorithm, it is the backbone algorithm rather than SACL that determines which NE it converges to, and SACL only makes this process faster.
>
> [4] Von Neumann, John, and Oskar Morgenstern. "Theory of games and economic behavior, 2nd rev." (1947).
>
> **Q8: How generalizable is SACL to other types of games such as cooperative or general-sum games?**
>
> SACL consists of three components: the subgame CL framework, the sampling metric, and the particle-based sampler. The framework and the sampler can be applied to cooperative and general-sum games because they don't require the zero-sum property. The only part to change is the metric.
> * For cooperative games, there are clear progression metrics like the success rate, so we can directly use these metrics as the sampling weight. In Appendix C.3, SACL achieves comparable results with one of the strongest CL algorithms for a cooperative game.
> * General-sum games are more complicated, and there is no clear metric to measure the subgames’ learning progress to the best of our knowledge. A possible way is to still start from Eq. (7) to derive a metric. It would be an interesting extension of SACL and we leave it for future work.
> ---
> We extend our sincere gratitude for your feedback and hope our answers have addressed your concerns. Your support is invaluable to us and we genuinely hope that our efforts merit a raise in your rating.

---

> > ### Comment · Reviewer_9JJp · 2023-08-20
> > **Thanks for the response**
> >
> > I want to thank the authors for their response. I find it helpful to address my questions. I would like to keep my original scores at this point.

---

### Official Review · Reviewer_z7mX · 2023-07-05

**Soundness:** 2 fair
**Presentation:** 3 good
**Contribution:** 2 fair
**Rating:** 5
**Confidence:** 3

**Summary:**

The paper proposes a subgame curriculum learning framework to accelerate multi-agent reinforcement learning (MARL) training for zero-sum games.  The framework adopts an adaptive initial state distribution by resetting agents to some previously visited states where they can quickly learn to improve performance. The paper derives a subgame selection metric that approximates the squared distance to Nash equilibrium (NE) values and further adopts a particle-based state sampler for subgame generation.  Experiments in the particle-world environment, Google Research Football environment, and hide-and-seek show that SACL produces much stronger policies than baselines.

**Strengths:**

- The paper provides a detailed analysis of the proposed approach and justifies it through experiments and analysis of a simple iterated Rock-Paper-Scissors game.
- The paper provides a clear and concise explanation of the proposed approach and its implementation.
- SACL is shown to produce much stronger policies than baselines in experiments conducted in the particle-world environment and Google Research Football environment.

**Weaknesses:**

- There is a gap between the motivated iterated Rock-Paper-Scissors game and the experiments conducted in the particle-world environment and Google Research Football. Since state and action space is discrete and finite in RPS, while these spaces are continuous in MPE and GRF. Also, The state is not communicative in RPS while it is communicative in MPE or GRF. In one rollout, two states are communicative if and only if each is reachable from the other with nonzero probability in a finite number of steps. So it is ok to use samples in RPS since it is a trade of time and space. While the state in MPE or GRF is something like position or speed, the advantage shown in the motivated scenario is hard to extend to more realistic environments.
- The link between eq.9 and eq.10 is weak, which should be the key to the theoretical contribution of the paper. I have several concerns about eq.10.
	- The estimated value function at different checkpoints represents different policies' values. Do the authors consider the missing $\pi$ inside the value function?
	- Does the asymptotic convergence of the estimated value function of a few sampled states mean the policies converge to NE? The explanation in Appendix is not theoretical and less convincing by giving examples.
- The subgame sampler is also essential, why the farthest point should be sampled first?
- Go is a good baseline for SACL. Since it is a fair game. To my understanding, SACL can be seen as an alternative to MCTS. One can start from the arbitrary state of a Go board, rollout, and then train the policy and value function. From this point of view, only having one rollout in Alg.2 might cause a large variance in the value function.

**Questions:**

See Weaknesses.

**Limitations:**

The paper does not explicitly mention any limitations of the proposed method. The paper has no potential negative societal impact.

---

> ### Author Rebuttal · Authors · 2023-08-09
>
> Thank you for your valuable feedback! We are heartened to see your recognition of our detailed analysis and strong experiment results. For your constructive questions, we hope the following response can address your concerns.
>
> **Q1: There is a gap between the iterated RPS game and the experiments in MPE and GRF, the advantage shown in the motivating example is hard to extend to more realistic environments.**
>
> We agree with the reviewer that the type of state/action space and the communicative property is different between the iterated RPS game and the MPE/GRF environment. However, the advantage of SACL is conceptually the same: starting from easier subgames and gradually moving to harder ones can solve the full game more efficiently.
>
> Use the MPE hard scenario as an example. Since the predators are in the top-right corner and the prey is in the bottom-left corner, the full game is hard to solve. A suitable order of subgames to learn would be to start from the easiest subgame where predators and prey are all near the center, and gradually move to harder subgames where they are at the edges or corners. This is conceptually the same as the case in the iterated RPS game where we learn from the last round to the first round. We also visualize the change of the prey’s initial position heatmap produced by SACL in the MPE hard scenario and it indeed starts from the center and moves to the edges and corners. Please see Fig. 1 in the global response PDF.
>
>
> **Q2: The estimated value function at different checkpoints represents different policies' values. Do the authors consider the missing $\pi$ inside the value function?**
>
> Yes, the value function $\tilde{V}_i^{(t)}$ represents the value of the current policy $\pi_i^{(t)}$, and a more clear way to write it would be $\tilde{V}_i^{\pi_i^{(t)}}$. We didn't explicitly write the policy $\pi$ in the value function for ease of notation. We will clarify this in our revised paper and we are sorry for the confusion caused.
>
>
> **Q3: Does the asymptotic convergence of the estimated value function of a few sampled states mean the policies converge to NE?**
>
> There might be some misunderstanding about the usage of Eq. (10). We are not using it as a criterion for NE convergence, but using it to prioritize subgames where the current policy is most likely to make improvement. If the weight $\tilde{w}(s)$ in Eq. (10) is small, it does not necessarily mean that the policy has converged in the subgame $MG(s)$. It is also possible that the subgame is too hard for the current policy and it is not making any progress. But in both cases, the induced subgame $MG(s)$ is not suitable for the current policy to learn and we should sample it less, which is exactly what a small weight will do.
>
>
> **Q4: Why the farthest point should be sampled first in the subgame sampler?**
>
> In the subgame sampler, we use a state buffer to approximate the whole state space and record the state weights. In principle, the states in the buffer should span the entire state space and distribute uniformly, but the rollout data is usually concentrated and very similar to each other. Therefore, we need to select states that are sufficiently far from each other to ensure good coverage of the state space.
>
> More formally, we want to select a subset $S'$ of size $K$ from the full set $S$, so that the sum of the shortest distances between states in subset $S'$ is maximized, i.e., $\max_{S'\subset S, |S'|=K}\sum_{s\in S'} \min_{s'\in S'}\|s-s'\|$. And farthest point sampling (FPS) is a greedy algorithm that efficiently finds an approximate optimal solution to this problem.
>
> The ablation study in Section 5.2 and Fig. 5 (d) also validates that FPS achieves the best result. We further visualize the state distribution in the buffer generated by different update methods in Fig. 2 of the global response PDF. Fig. 2(a-c) show the heatmaps of the predators’ position. Fig. 2(d-f) run PCA on the full state space and show the projection of the states in the buffer to the two-dimensional space. The results shwo that if we randomly select states or greedily select states with high weights, the states in the buffer can become very concentrated and can't approximate the whole state space.
>
>
> **Q5: Only having one rollout in Alg. 2 might cause a large variance in the value function.**
>
> In the actual implementation, we run hundreds or thousands of rollouts and train on a batch of collected data. We will make the following changes to Alg. 2 to make it clear:
> >...
> >
> > For each parallel environment:
> >
> > &nbsp;&nbsp;&nbsp;&nbsp;Sample $s^0 \sim \text{sampler}(M)$ with probability $p$, else $s^0\sim\rho(\cdot)$;
> >
> > &nbsp;&nbsp;&nbsp;&nbsp;Rollout in $MG(s^0)$ and collect samples;
> >
> > Train $\{\pi_i, V_i\}_{i=1}^2$ on the samples via MARL;
> >
> > ...
>
> ---
>
> We would like to express our appreciation for your constructive review. We have carefully addressed each of your concerns and believe that our responses demonstrate the value of our proposed approach. We kindly request you reconsider the rating for our submission and genuinely hope our efforts will warrant a higher evaluation.

---

### Official Review · Reviewer_dzKo · 2023-07-12

**Soundness:** 2 fair
**Presentation:** 4 excellent
**Contribution:** 2 fair
**Rating:** 5
**Confidence:** 4

**Summary:**

This paper presents an algorithm (SACL) for accelerating MARL training in zero-sum Markov games based on the subgame curriculum learning framework. A sampling metric based on approximated squared distance to NE and a particle-based sampler are proposed to sample states for subgame generation. Experiment results show the superiority of SACL in producing stronger policies and boosting training efficiency.

**Strengths:**

1. The paper is well-written and easy to follow. The general idea of the algorithm is well illustrated via a toy example: the iterated RPS games.
2. The experiment results show both the efficiency and effectivity of SACL.


**Weaknesses:**

* I am not sure the game considered in this paper is perfect-information or imperfect-information. From the Rock-Paper-Scissor example and the baseline methods considered (NeuRD and PSRO), I am aussming the case of imperfect-information.  If it is the case of imperfect-information, the optimal policy for a subgame depends on the distribution of hidden information. Yet, I don't see any component of SACL that deals with this distribution.  For instance, the DeepStack (a poker AI) algorithm uses the agent's range and opponent counterfactual values when resolving a subgame in poker.


* Also, the choice of the metric and oracle sampler appears ad-hoc to me.
  1. Why can we  use Equ 10 to approximate Equ 9? Is this estimator unbiased or with low variance?
  2. In Equ 10, is the algorithm stable for different choices of \alpha? It would be good to see some discussion or ablation study on it.


**Questions:**

 How to define the distance between two points when using FPS to select the points?

**Limitations:**

Please see weaknesses.

---

> ### Author Rebuttal · Authors · 2023-08-09
>
> Thank you for your constructive comments and questions! We are encouraged to see your positive assessment of SACL’s efficiency and effectiveness. And we hope the following responses can address your concerns.
>
> **Q1: Is the game considered in this paper perfect-information or imperfect-information?**
>
> In this paper, we use the formulation of Markov games, which is different from the formulation of extensive games with perfect-information or imperfect-information. SACL can be directly used in fully-observable Markov games where the states contain all information about the game and are observable to all agents. For partially-observable Markov games, though some of the information is hidden from the agents, the states still contain all information of the game and it is also possible to run SACL in these games. As the reviewer said, an important part is to deal with the distribution of hidden information. A way to do that is to replace states in prioritized sampling with infosets, i.e., sets of states that are indistinguishable to agents, and maintain the distribution of states within each infoset. In each episode, we first use prioritized sampling to select an infoset and then sample a state from the infoset to generate the subgame. In this way, we keep the distribution of hidden information and also build a subgame curriculum to accelerate training.
>
>
> **Q2: Why can we use Eq. (10) to approximate Eq. (9)?**
>
> We put the discussion about Eq. (9) and (10) in Appendix A.3 due to the limited space in the main text. In short, Eq. (10) does not directly approximate Eq. (9) but estimates the squared distance to the next local optimal value. As training continues, the final optimal value is the NE value and Eq. (10) approximates Eq. (9).
>
> Consider a simple case where the value function changes monotonically throughout training, as shown in Fig. 6 (in Appendix A.3), the first term in Eq. (10) can be regarded as a first-order approximation of the bias term in Eq. (9). However, in zero-sum games, the value function may oscillate up and down in different stages (like in hide-and-seek). In this case, Eq. (10) becomes the approximated squared difference between the current value and the next local optimal value, which is derived in Eq. (12) and shown in Fig. 7. Therefore, by using the weight in Eq. (10), we are not directly prioritizing states whose values are far from the NE value, but states whose values are far from the local optimal value of the current stage. And by accelerating the learning in each stage, we make the NE learning process more efficient in total.
>
>
> **Q3: Is the algorithm stable for different choices of $\alpha$ in Eq. (10)?**
>
> Yes, SACL is stable for different $\alpha$. The ablation study on $\alpha$ can be found in Appendix C.4 and Fig. 20(d). We test different values from {$0.3, 0.7, 1.0, 3.0$} on the MPE hard scenario and they achieve comparable results. This shows that our algorithm is not sensitive to the hyperparameter $\alpha$.
>
>
> **Q4: How to define the distance between two points when using FPS to select the points?**
>
> Because different dimensions of the state vector (or point) may have different value ranges and affect the distance calculation, we first normalize each dimension of the state vector so that their ranges are all $[0, 1]$. Then the distance between two states is simply the Euclidean distance. The distance between a state $s_a$ and a set of states $S$ is the smallest distance between $s_a$ and any state in $S$, i.e., $\text{min}_{s\in S} \|s_a - s\|$.
>
> ---
>
> We greatly appreciate your thorough review of our work and hope our answers effectively address your concerns. We would be very grateful if you could re-evaluate our paper based on the responses and consider raising the rating for our submission.

---

> > ### Comment · Reviewer_dzKo · 2023-08-14
> >
> > Thanks very much for your rebuttal. After reading your response to **Q1**, I am still deeply concerned with the way SACL constructs a subgame for imperfect-information games.
> >
> > - Subgame solving done for imperfect-information games is fundamentally different from that for perfect-information games [1-4].
> >
> > - For imperfect-information games (e.g., poker), there has been extensive literature on subgame solving (how to construct&solve a subgame) . The paper barely touches these methods, only citing [1] without any discussion. Given the existing literature on how subgame solving is currently being done [1-4], the subgame generation, sampling, and solving in SACL **look rather ad hoc to me**. Without any theoretical result or experimental comparison to any of [1-4] on `default'  benchmarks such as poker, I am not convinced that the subgame curriculum learning method presented in this paper is a solid contribution to the general subgame solving research community.
> >
> > [1] Brown, Noam, and Tuomas Sandholm. "Safe and nested subgame solving for imperfect-information games." Advances in neural information processing systems 30 (2017).
> >
> > [2] Zhang, Brian, and Tuomas Sandholm. "Subgame solving without common knowledge." Advances in Neural Information Processing Systems 34 (2021): 23993-24004.
> >
> > [3] Burch, Neil, Michael Johanson, and Michael Bowling. "Solving imperfect information games using decomposition." Proceedings of the AAAI Conference on Artificial Intelligence. Vol. 28. No. 1. 2014.
> >
> > [4] Moravcik, Matej, et al. "Refining subgames in large imperfect information games." Proceedings of the AAAI Conference on Artificial Intelligence. Vol. 30. No. 1. 2016.
> >
> > ...

---

> > > ### Author Response · Authors · 2023-08-16
> > > **Reply to Reviewer dzKo**
> > >
> > > Thank you for your comments. We would like to clarify some aspects of our work that may not have been fully grasped.
> > >
> > > In this work, we use SACL to accelerate training in fully-observable Markov games. Please note that Markov games (where the agents make decisions simultaneously in each step) and extensive-form games are different formulations, and a fully-observable Markov game can be an imperfect-information extensive-form game, e.g., the iterated RPS game in Section 3. The subgame of a fully-observable Markov game can be solved by standard MARL methods like Minimax-Q. In the previous rebuttal, we discussed a potential way to extend SACL to partially-observable Markov games, and we agree with the reviewer that it could be further improved by methods like subgame solving. However, it is out of the scope of this paper and we leave this extension of SACL for future work.
> > >
> > > In addition, we would like to respectfully point out that subgame (re)solving techniques in [1-4] are fundamentally different from ours. The idea of subgame solving is to first get a blueprint strategy of the abstracted game and use it to play the original game. As the game progresses and the remaining game becomes tractable, the specific subgame is solved in real-time to create a combined final policy. Subgame solving is **applied online during the game** when extra time and computing resources are available and considers **“how to solve the subgame”**. By contrast, SACL is used to learn a policy **before playing the game** and accelerate the training by generating an appropriate order of subgames to learn, i.e., we focus on **“which subgame to solve”**.  These two methods exhibit distinct differences, and we have full confidence that SACL makes a solid and valuable contribution to the community. Your suggestion to add more discussions on subgame solving is well-received and we will make sure the clarification and additional references are included in our revised paper.
> > >
> > > We are committed to making the necessary changes to clarify all aspects. We hope these explanations and clarifications can help you have a more thorough understanding and a better evaluation of our paper.

---

> > > > ### Comment · Reviewer_dzKo · 2023-08-16
> > > >
> > > > Thanks very much for the reply. I just would like to respond that:
> > > > - Although [1-4] is applied online during the game, I don't see any reason why they can not be used offline before playing the game for your purposes.
> > > > - Subgame solving in your case is not solely about “which subgame to solve”, it also involves how to construct a subgame and how the subgame solving result relates to the full game (because eventually we would like to have a good strategy for the full game). From this perspective, I think  [1-4] are closely related to SACL, and the current version of SACL pays little attention to previous subgame solving literature such as [1-4] (both in related work and experimental study).
> > > > - Given that there is not any **theoretical result** demonstrating that the subgame solving technique in SACL makes sense,  the comparison with [1-4] appears necessary for me to acknowledge the contribution of SACL.

---

> > > > > ### Author Response · Authors · 2023-08-17
> > > > > **Reply to Reviewer dzKo**
> > > > >
> > > > > Thank you for the comments. For your first question, there are a few points that we would like to clarify.
> > > > >
> > > > > 1. We would like to emphasize that the goal of this work is to accelerate learning in complex fully-observable Markov games, and the iterated RPS game is a simplified example only for demonstration purposes. In our experiments, the environments are all complex games that have high-dimensional continuous state space and long horizons, and the learning agents do not know the transition of the games (following the standard assumption in reinforcement learning).
> > > > > 2. Subgame solving typically uses iterative updates based on regret matching to find the policy, which requires the traverse of the game tree. It is possible to do so if we know the game structure and the subgame is tractable, like in poker. But in our setting, we consider long-horizon and even infinite-horizon Markov games with unknown transitions, it is impossible to traverse the entire game tree and use regret-matching methods.
> > > > > 3. Subgame solving is designed for online computation because it only needs to solve the subgame induced by agents’ prior actions, which is usually much more tractable than the full game. For example, in poker, after the first few cards are dealt, the resulting subgame is much smaller. Applying subgame solving to offline settings would require solving every single subgame in the game tree to find a policy for the entire game, which is not helpful in terms of computational efficiency. In fact, in our setting, given the high-dimensional continuous state space, there can be infinite subgames and it is impossible to solve all of them. In contrast, our work follows the typical multi-agent reinforcement learning paradigm to train a neural network based policy. And by strategically sampling subgames to train the network, we get a policy that is generalizable to different states in the Markov games.
> > > > >
> > > > > To summarize, the existing subgame solving techniques are not directly applicable to our setting as we focus on accelerating learning in complex Markov games with agents having no knowledge of the game transitions. And we agree with the reviewer that there is a potential to combine the existing subgame solving methods with multi-agent reinforcement learning to solve large-scale extensive-form games when the game structure is known a priori.
> > > > >
> > > > > Your second and third questions have been well explained in our paper under the setting we are considering, i.e., fully-observable Markov games. We are willing to provide references and further discussions.
> > > > >
> > > > > 1. How to construct a subgame: L74-77 "For infinite-horizon Markov games, a subgame $MG(s)$ is defined as the Markov game induced by starting from state $s$, i.e., $\rho(s)=1$...". The state contains all information about the game's current status, and the induced subgame is constructed by resetting the environment to this state.
> > > > > 2. How the subgame solving result relates to the full game: Since the game is Markovian and fully observable, the NE policy of a subgame is determined only by itself, and does not depend on other unreached subgames. Therefore, the subgame solving result can be directly used as part of the full game's NE policy without any change. This relationship is shown in Section 3.2 where we use subgame curriculum learning to solve the iterated RPS game. We will add an explicit explanation to make it more clear.
> > > > > 3. Theoretical result demonstrating the subgame solving technique in SACL makes sense: SACL can use any MARL algorithm to solve the subgames and we do not consider the theoretical property of the method used. But we do prove in Proposition 1 (L164-166) that if the backbone MARL algorithm is guaranteed to converge to an NE in zero-sum Markov games, subgame curriculum learning with the MARL method also produces an NE of the original Markov game. This shows that SACL does not change the convergence property of the MARL methods used and it does make sense.
> > > > >
> > > > > We have carefully addressed each of your concerns and believe that our responses demonstrate the value of our proposed approach. We genuinely hope our efforts will lead to a higher evaluation.

---

> > > > > > ### Comment · Reviewer_dzKo · 2023-08-17
> > > > > >
> > > > > > Thanks very much for your detailed response. I will update my score accordingly.

---

### Official Review · Reviewer_7PBD · 2023-07-12

**Soundness:** 3 good
**Presentation:** 3 good
**Contribution:** 2 fair
**Rating:** 5
**Confidence:** 2

**Summary:**

This paper proposes a framework for learning Nash equilibria in zero-sum Markov games based on subgame curriculum learning. Novel sampling metrics for subgames generation are proposed. The proposed SACL algorithm is able to achieve equal performance at lower  sample complexity compared with self-play algorithms.

**Strengths:**

- Proposed a novel framework for learning NE in zero-sum games. This can be combined with different RL algorithms to produce learning mechanisms with lower sample complexity.
- The presentation of the paper is clear.

**Weaknesses:**

- While the baseline methods require an exponential complexity, the proposed SACL costs approximately half, which is a limited practical improvement especially considering real-world applications as the authors suggested.

**Questions:**

- Are there potential ways to enlarge the complexity improvement? This could significantly improve the impact of the proposed method.

I am not familiar with this field.

**Limitations:**

A more thorough discussion of the limitations and future steps could be beneficial.

---

> ### Author Rebuttal · Authors · 2023-08-09
>
> Thank you for your review and feedback! We appreciate your recognition of the novelty of our work and the clarity of our presentation. In response to your questions, we provide the following explanation.
>
> **Q1: While the baseline methods require an exponential complexity, the proposed SACL costs approximately half, which is a limited practical improvement especially considering real-world applications as the authors suggested.**
>
> We would like to clarify that SACL substantially accelerates the learning process instead of just costing half the samples as shown in both the motivating example and in our experiments.
>
> More concretely, in the motivating example named iterated RPS game, we show both theoretically and empirically that SACL greatly reduces the sample complexity from exponential to linear.
>
> In our experiments, SACL learns much stronger policies with the lowest exploitability in both MPE and GRF. The learning curve shows that SACL achieves this with much less sample complexity. Take the MPE hard results in Fig. 4(b) for an example, SACL converges to an approximate NE with about $1.5 \times 1e3$ exploitability using 40M samples. We trained SP for 400M samples and it still couldn't reach the same exploitability, which shows that SACL is at least $10$ times faster than the best baseline.
>
> In the complex HnS environment, as the reviewer mentioned, SACL uses about half the samples. This is NOT a trivial improvement in practice considering the long training time. Using a single A100 GPU, it takes about $10$ days to train the policy with MAPPO. With half the samples, SACL only uses about $5-6$ days. For more complex real-world applications like OpenAI Five which is trained for $10$ months, half the samples could save a substantial amount of time.
>
> To support our argument, we would like to quote a few lines from the other reviewers and hope they can help the reviewer have a more comprehensive understanding of our work.
> * Reviewer 9JJp: “The paper addresses an important and challenging problem of reducing the computational cost of solving complex zero-sum games with MARL, which has many potential applications and implications.”
> * Reviewer z7mX: “SACL is shown to produce much stronger policies than baselines in experiments conducted in the particle-world environment and Google Research Football environment.”
> * Reviewer dzKo: “Experiment results show the superiority of SACL in producing stronger policies and boosting training efficiency.”
>
>
> **Q2: Are there potential ways to enlarge the complexity improvement? This could significantly improve the impact of the proposed method.**
>
> As discussed in the previous question, our analysis and experiments show SACL already greatly reduces the complexity and accelerates NE learning in zero-sum games. We believe SACL can be helpful to make MARL training in complex zero-sum games more affordable to the community.
>
> Furthermore, since SACL is a general framework that does not require domain knowledge of the game, it is possible to further accelerate training by incorporating domain-specific design. For example, the subgame complexity in MPE is determined largely by the position of agents and is less affected by the location of obstacles. So when we use FPS to select representative states as described in Section 4.3, we can put more weight on the dimension for agents' position and less on the landmarks'. This domain-specific technique can further accelerate NE learning in MPE.
>
> ---
>
> We genuinely value your feedback and hope our answers can help you better evaluate the contribution of our work. We kindly request your reconsideration of the paper’s rating based on our responses and hope to receive your support for its acceptance.

---

> > ### Comment · Reviewer_7PBD · 2023-08-20
> > **Thank you**
> >
> > I thank the authors for their detailed response, which has helped me better appreciate the paper's contributions. I am raising the score to 5.

---

### Author Rebuttal · Authors · 2023-08-09

We would like to thank all reviewers for taking the time to review our submission and providing constructive comments on our work. We are heartened by the consensus among reviewers about the strengths of our work, which align with our intentions and efforts:

1. **Novelty and significance:** We appreciate that all reviewers agree our proposed approach is novel and substantially accelerates NE learning in zero-sum games. It's encouraging to see the comment by reviewer 9JJp that "the paper addresses an important and challenging problem of reducing the computational cost of solving complex zero-sum games with MARL, which has many potential applications and implications."
2. **Thorough evaluation:** We are pleased to see the reviewers' acknowledgment of our comprehensive experiments and evaluations in three different games. As mentioned by reviewer dzKo, "The experiment results show both the efficiency and effectivity of SACL."
3. **Clarity and presentation:** We are gratified to see that our efforts to present our ideas clearly have been well-received by all reviewers. Many reviewers mentioned the illustrating example and reviewer z7mX said "The paper provides a detailed analysis of the proposed approach and justifies it through experiments and analysis of a simple iterated Rock-Paper-Scissors game."

In response to the specific concerns and suggestions raised by each reviewer, we provide a detailed discussion for each of your reviews. We have also prepared a PDF that contains the additional experiment results. We believe these figures will provide a more comprehensive visualization of our results and further reinforce the validity of our work.

Best regards,

Submission 6466 Authors

---

### Author Response · Authors · 2023-08-19
**A Gentle Reminder**

Dear AC and Reviewers,

Thank you once again for your insightful comments and valuable guidance. Your feedback has played a crucial role in enhancing both the quality and clarity of our paper.

While we have been actively engaged in the discussion phase for several days, we have not yet received all the anticipated further responses.

We remain eager to delve into the strengths and merits of our work with you. Should you have any additional questions or require further clarification, please do not hesitate to reach out. Your insights and suggestions are not only highly appreciated but integral to our process, and we stand ready to make any necessary improvements to the paper.

Best Regards,

Submission 6466 Authors

---

### Decision · Program_Chairs · 2023-09-21

**Decision:**

Reject

**Comment:**

The paper proposed a novel subgame curriculum learning framework for zero-sum games. It adopts an adaptive initial state distribution by resetting agents to some previously visited states where they can quickly learn to improve performance. During the review period, the author had multiple rounds of effective communication with the reviewers, in which the author addressed some of the reviewers' concerns and received recognition from the reviewers. However, the reviewers' attitude towards the scientific innovation of this paper is conservative. Combining the current manuscript and rebuttal content, I suggest that the author can state several core contributions of this paper more clearly (the current Introduction is confusing in my opinion), especially some core formulas (such as eq. 9 and eq.10). The derivation of should be clear, and some other derivation should be provided with sufficient appendices to aid reading if necessary. Overall, this paper does not meet the criteria for acceptance by NeurIPS. I strongly encourage the authors to address the comments from this round and submit to the incoming ML conferences.